# REVISITING GROUP ROBUSTNESS:
# CLASS-SPECIFIC SCALING IS ALL YOU NEED

## ABSTRACT

Group distributionally robust optimization, which aims to improve robust accuracies such as worst-group or unbiased accuracy, is one of the mainstream algorithms to mitigate spurious correlation and reduce dataset bias. While existing approaches have apparently gained performance in robust accuracy, these improvements mainly come from a trade-off at the expense of average accuracy. To address the challenges, we first propose a simple class-specific scaling strategy to control the trade-off between robust and average accuracies flexibly and efficiently, which is directly applicable to existing debiasing algorithms without additional training; it reveals that a naïve ERM baseline matches or even outperforms the recent debiasing approaches by adopting the class-specific scaling. Then, we employ this technique to 1) evaluate the performance of existing algorithms in a comprehensive manner by introducing a novel unified metric that summarizes the trade-off between the two accuracies as a scalar value and 2) develop an instance-wise adaptive scaling technique for overcoming the trade-off and improving the performance even further in terms of both accuracies. Experimental results verify the effectiveness of the proposed frameworks in both tasks.

## 1 INTRODUCTION

Machine learning models have achieved remarkable performance in various tasks via empirical risk minimization (ERM). However, they often suffer from spurious correlation and dataset bias, failing to learn proper knowledge about minority groups despite their high overall accuracies. For instance, because digits and foreground colors have a strong correlation in the colored MNIST dataset (Arjovsky et al., 2019; Bahng et al., 2020), a trained model learns unintended patterns of input images and performs poorly in classifying the digits in minority groups, in other words, when the colors of the digits are rare in the training dataset.

Since spurious correlation leads to poor generalization performance in minority groups, group distributionally robust optimization (Sagawa et al., 2020) has been widely studied in the literature about algorithmic bias. Numerous approaches (Huang et al., 2016; Sagawa et al., 2020; Seo et al., 2022a; Nam et al., 2020; Sohoni et al., 2020; Levy et al., 2020; Liu et al., 2021) have presented high robust accuracies such as worst-group or unbiased accuracies in a variety of tasks and datasets, but, although they clearly sacrifice the average accuracy, comprehensive evaluation jointly with average accuracy has not been actively explored yet. Refer to Figure 1 about the existing trade-offs of algorithms.

This paper addresses the limitations of the current research trends and starts with introducing a simple post-processing technique, *robust scaling*, which efficiently performs class-specific scaling on prediction scores and conveniently controls the trade-off between robust and average accuracies. It allows us to identify any desired performance points, *e.g.*, for average accuracy, unbiased accuracy, worst-group accuracy, or balanced accuracy, on the accuracy trade-off curve using a single model with marginal computational overhead. The proposed robust-scaling method can be easily plugged into various existing debiasing algorithms to improve the desired target objectives within the trade-off. One interesting observation is that, by adopting the proposed robust scaling, even the ERM baseline accomplishes competitive performance compared to the recent group distributionally robust optimization approaches (Liu et al., 2021; Nam et al., 2020; Sagawa et al., 2020; Kim et al., 2022; Seo et al., 2022a; Creager et al., 2021; Levy et al., 2020; Kirichenko et al., 2022; Zhang et al., 2022)

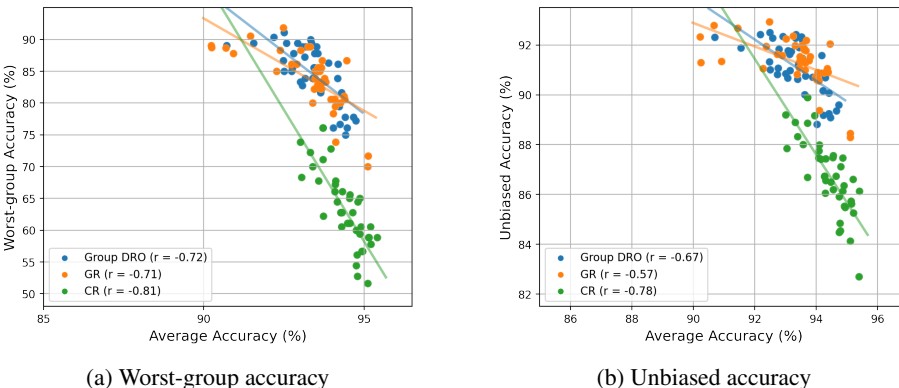

(a) Worst-group accuracy          (b) Unbiased accuracy

Figure 1: The scatter plots that illustrate trade-off between robust and average accuracies on CelebA dataset using ResNet-18. We visualize the results from multiple runs of each algorithm and present the relationship between the two kinds of accuracies. The lines denote the linear regression results of individual algorithms and $r$ in the legend indicates its Pearson coefficient correlation, which validates the strong negative correlation between both accuracies.

without extra training, as illustrated in Figure 2. We will present the results from other debiasing algorithms in the experiment section.

By taking advantage of the robust scaling technique, we develop a novel comprehensive evaluation metric that consolidates the trade-off of the algorithms for group robustness, leading to a unique perspective of group distributionally robust optimization. To this end, we first argue that comparing the robust accuracy without considering the average accuracy is incomplete and a unified evaluation of debiasing algorithms is required. For a comprehensive performance evaluation, we introduce a convenient measurement referred to as *robust coverage*, which considers the trade-off between average and robust accuracies from the Pareto optimal perspective and summarizes the performance of each algorithm with a scalar value. Furthermore, we propose a more advanced robust scaling algorithm that applies the robust scaling to each example adaptively based on its cluster membership at test time to maximize performance. Our instance-wise adaptive scaling strategy is effective to overcome the trade-off between robust and average accuracies and achieve further performance gains in terms of both accuracies.

**Contribution.** We present a simple but effective approach for group robustness by the analysis of trade-off between robust and average accuracies. Our framework captures the full landscape of robust-average accuracy trade-offs, facilitates understanding the behavior of existing debiasing techniques, and provides a way for optimizing the arbitrary objectives along the trade-off using a single model without extra training. Our main contributions are summarized as follows.

- We propose a training-free class-specific scaling strategy to capture and control the trade-off between robust and average accuracy with marginal computational cost. This approach allows us to optimize a debiasing algorithm for arbitrary objectives within the trade-off.

- We introduce a novel comprehensive performance evaluation metric via the robust scaling that summarizes the trade-off between robust and average accuracies as a scalar value from the Pareto optimal perspective.

- We develop an instance-wise robust scaling algorithm by extending the original class-specific scaling with joint consideration of feature clusters. This technique is effective to overcome the trade-off and improve both robust and average accuracy further.

- The extensive experiments analyze the characteristics of existing methods and validate the effectiveness of our frameworks on the multiple standard benchmarks.

## 2 RELATED WORKS

Mitigating spurious correlation has been emerged as an important problem in a variety of areas in machine learning. Sagawa *et al.* (Sagawa et al., 2020) propose group distributionally robust opti-

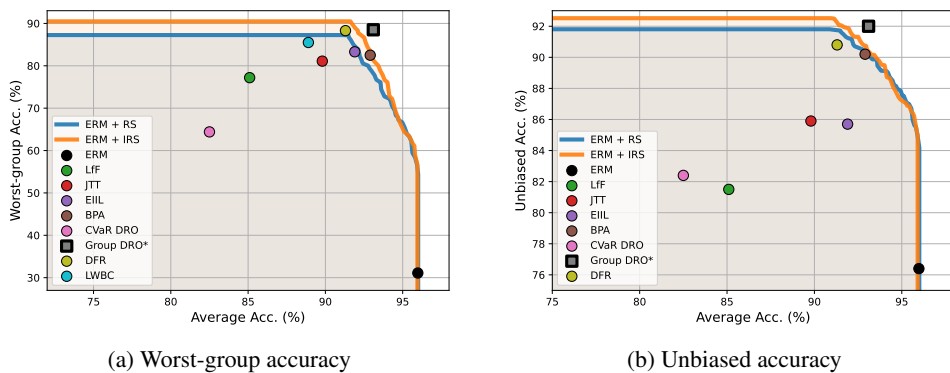

(a) Worst-group accuracy          (b) Unbiased accuracy

Figure 2: Comparison between the baseline ERM and existing debiasing approaches on CelebA dataset using ResNet-50. Existing works have achieved meaningful improvements in robust accuracy over ERM, but our robust scaling strategies (RS, IRS) enable ERM to catch up or even outperform them without any training.

mization (Group DRO) and provide a practical assumption; training examples are given in groups, and arbitrary test distribution is represented by a mixture of these groups. Because models which rely on the spurious correlation yield poor worst-group performance, group distributionally robust optimization, which aims to maximize worst-group accuracy, is widely adopted to deal with spurious correlation, and the methods can be mainly categorized into three folds as follows.

**Sample Reweighting** The most popular approaches are assigning different training weights to each samples to focus on the minority groups, where the weights are based on the group frequency or loss. Group DRO (Sagawa et al., 2020) minimizes the worst-group loss by reweighting based on the average loss per group. Although Group DRO achieves robust results against group distribution shifts, it requires training examples with group supervision. To handle this limitation, several unsupervised approaches have been proposed that do not exploit group annotations. George (Sohoni et al., 2020) and BPA (Seo et al., 2022a) extend Group DRO in an unsupervised way, where they first train the ERM model and use this model to infer pseudo-groups via feature clustering. CVaR DRO (Levy et al., 2020) minimizes the worst loss over all $\alpha$-sized subpopulations, which upperbounds the worst-group loss over the unknown groups. LfF (Nam et al., 2020) simultaneously trains two models, one is with generalized cross-entropy and the other one is with standard cross-entropy loss, and reweights the examples based on their relative difficulty score. JTT (Liu et al., 2021) conducts a two-stage procedure, which upweights the examples that are misclassified by the first-stage model. Idrissi et al. (2022) analyze simple data subsampling and reweighting baselines based on group or class frequency to handle dataset imbalance issues. LWBC (Kim et al., 2022) employs an auxiliary module to identify bias-conflicted data and assigns large weights to them.

**Representation Learning** On the other hand, some approaches aim to learn debiased representations directly to avoid spurious correlation. ReBias (Bahng et al., 2020) adopts Hilbert-Schmidt independence criterion (Gretton et al., 2005) to learn feature representations independent of the predefined biased representations. Cobias (Seo et al., 2022b) defines conditional mutual information between feature representations and group labels as a bias measurement, and employ it as a debiasing regularizer. IRM (Arjovsky et al., 2019) learns invariant representations to diverse environments, where the environment variable can be interpreted as the group. While IRM requires the supervision of the environment variable, EIIL (Creager et al., 2021) and PGI (Ahmed et al., 2020) are the unsupervised counterparts, which assign each training example to the environment that violates the IRM objective maximally.

**Post-processing** While most previous studies are in-processing approaches that perform feature representation learning or sample reweighting during training to improve group robustness, our framework does not fall into those categories; it deals with group robust optimization by simple post-processing with class-specific score scaling, which does not require any additional training. Similar to our method, some post-processing techniques, including temperature (Guo et al., 2017) or Platt (Platt, 2000) scaling, are widely exploited in the literature of confidence calibration, but it is

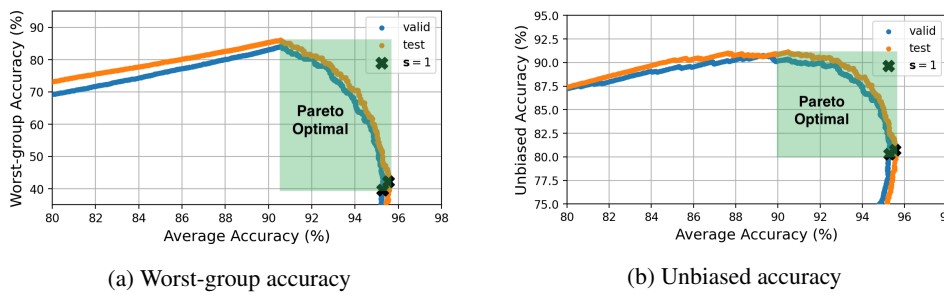

(a) Worst-group accuracy             (b) Unbiased accuracy

Figure 3: The relation between the robust and average accuracies obtained by varying the class-specific scaling factor with ERM model on the CelebA dataset. The black marker denotes the original point, where the scaling is not applied. The points, located within the green area, on the trade-off curve represent the Pareto optimums.

not applicable in our task because it scales the prediction scores equally for each class and does not change the label predictions. Recently, DFR (Kirichenko et al., 2022) demonstrated that retraining only the last layer of the model with group-balanced dataset can match the performance of existing approaches, but they still require additional training different from our framework.

## 3 FRAMEWORK

This section first presents a class-specific scaling technique, which captures the landscape of the trade-off and identifies the optimal performance points for desired objectives along the trade-off curve. Based on the scaling strategy, we introduce a novel unified metric for evaluating the group robustness of an algorithm with consideration of the trade-off. Finally, we propose an instance-wise class-specific scaling approach to overcome the trade-off and further improve the performance.

### 3.1 PROBLEM SETUP

Let us consider a triplet $(x, y, a)$ consisting of an input feature $x \in \mathcal{X}$, a target label $y \in \mathcal{Y}$, and an attribute $a \in \mathcal{A}$. We construct a group based on a pair of a target label and an attribute, $g := (y, a) \in \mathcal{Y} \times \mathcal{A} =: \mathcal{G}$. We are given $n$ training examples without attribute annotations, e.g., $\{(x_1, y_1), ..., (x_n, y_n)\}$, while $m$ validation examples have group annotations for model selection, e.g., $\{(x_1, g_1), ..., (x_m, g_m)\} = \{(x_1, y_1, a_1), ..., (x_m, y_m, a_m)\}$.

Our goal is to learn a model $f_\theta(\cdot) : \mathcal{X} \to \mathcal{Y}$ that is robust to group distribution shifts. To measure the group robustness, we typically refer to the robust accuracy such as unbiased accuracy (UA) and worst-group accuracy (WA). The definitions of UA and WA require the group-wise accuracy (GA), which is given by

$$\text{GA}_{(r)} := \frac{\sum_i \mathbb{1}(f_\theta(\mathbf{x}_i) = y_i, g_i = r)}{\sum_i \mathbb{1}(g_i = r)}, \tag{1}$$

where $\mathbb{1}(\cdot)$ denotes an indicator function and $\text{GA}_{(r)}$ is the accuracy of the $r^{\text{th}}$ group samples. Then, the robust accuracies are defined by

$$\text{UA} := \frac{1}{|\mathcal{G}|} \sum_{r \in \mathcal{G}} \text{GA}_{(r)} \quad \text{and} \quad \text{WA} := \min_{r \in \mathcal{G}} \text{GA}_{(r)}. \tag{2}$$

The goal of the group robust optimization is to ensure robust performance by improving UA and WA regardless of the group membership of a sample.

### 3.2 ROBUST SCALING: CLASS-SPECIFIC SCALING FOR GROUP ROBUSTNESS

We perform a simple non-uniform class-specific scaling of the scores corresponding to individual class and potentially change the final decision of the classifier to achieve the robustness by improving

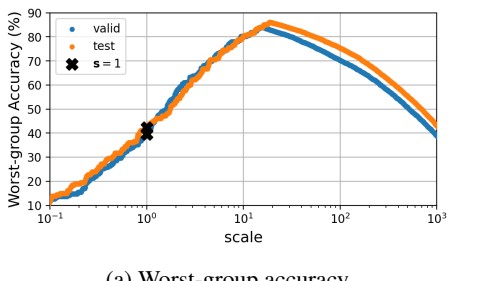
(a) Worst-group accuracy

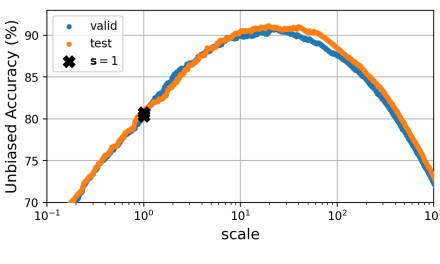
(b) Unbiased accuracy

Figure 4: Effects of varying the class-specific scaling factors on the robust accuracy using ERM model on the CelebA dataset. Since this experiment is based on the binary classifier, a single scaling factor is varied with the other fixed to one. This shows that the optimal scaling factor identified in the validation set can be used in the test set to get the final robust prediction.

worst-group or unbiased accuracies while sacrificing the average accuracy. Formally, the prediction with the class-specific scaling is given by

$$\arg\max_c \ (\mathbf{s} \odot \hat{\mathbf{y}})_c, \tag{3}$$

where $\hat{\mathbf{y}} \in \mathbb{R}^C$ is a prediction score vector, $C$ is the number of classes, $\mathbf{s} \in \mathbb{R}^C$ is a $C$-dimensional scaling coefficient vector, and $\odot$ denotes the element-wise product operator.

Based on the ERM model, we obtain a set of the average and robust accuracy pairs using a wide range of the class-specific scaling factors and illustrate their relations in Figure 3. The black markers indicate the point with a uniform scaling, *i.e.*, $\mathbf{s} = (1, \dots, 1) \in \mathbb{R}^C$. The graphs show that a simple class-specific scaling effectively captures the full landscape of the trade-off of the two accuracies. We also visualize the relationship between scaling factors and robust accuracies in Figure 4, where the curves constructed based on validation and test splits are sufficiently well-aligned to each other. This validates that we can identify the desired optimal points between robust and average accuracies in the test set by following a simple strategy: 1) finding the optimal class-specific scaling factors that maximize the target objective (UA, WA, or AA) in the validation set, and 2) apply the scaling factors to the test set. We refer to this scaling strategy for robust prediction as *robust scaling*.

To find the optimal scaling factor $\mathbf{s}$, we adopt a greedy search[1] and the entire process takes less than a second. Note that the robust scaling is a post-processing method, so it can be easily applied to any kinds of existing robust optimization methods without extra training. Moreover, our method can find any desired performance points on the trade-off envelop using a single model. For example, there may be scenarios in which multiple objectives are required to solve a problem, but the robust scaling approach is flexible enough to handle the situation as we only need to apply a robust scaling for each target metric using a single model. Meanwhile, other existing robust optimization methods have limited flexibility and require training of separate models for each target objective.

### 3.3 ROBUST COVERAGE FOR COMPREHENSIVE PERFORMANCE EVALUATION

Although the robust scaling identifies the desired performance point on the trade-off curve, from the perspective of performance evaluation, it still reflects only a single point on the trade-off curve while ignoring all other possible Pareto optimums. For a more comprehensive evaluation of an algorithm, we propose a convenient evaluation metric that yields a scalar summary of the robust-average accuracy trade-off. Formally, we define the *robust coverage* as

$$\text{Coverage} := \int_{c=0}^{1} \max_{\mathbf{s}} \left\{ \text{WA}^{\mathbf{s}} | \text{AA}^{\mathbf{s}} \geq c \right\} dc \approx \sum_{d=0}^{D-1} \frac{1}{D} \max_{\mathbf{s}} \left\{ \text{WA}^{\mathbf{s}} | \text{AA}^{\mathbf{s}} \geq \frac{d}{D} \right\}, \tag{4}$$

where $\text{WA}^{\mathbf{s}}$ and $\text{AA}^{\mathbf{s}}$ denote the worst-group and average accuracies given by the scaled prediction using (3) with the scaling factor $\mathbf{s}$, and $D = 1000$ is the number of slices for discretization. The

---

[1]We search for the scaling factor of each class in a greedy manner. Specifically, we first find the best scaling factor for a class and then determine the optimal factors of the remaining classes sequentially conditioned on the previously estimated ones.

Table 1: Experimental results of the robust scaling (RS) on the CelebA dataset using ResNet-18 with the average of three runs (standard deviations in parenthesis), where RS is applied to maximize each target metric independently. On top of all existing approaches, RS can maximize all target metrics consistently. *Gain* indicates the average (standard deviations) of performance improvement of RS for each run.

| Method | Group Supervision | Robust Coverage | | Accuracy (%) | | |
|---|---|---|---|---|---|---|
| | | Worst-group | Unbiased | Worst-group | Unbiased | Average |
| ERM | | - | - | 34.5(6.1) | 77.7(1.8) | 95.5(0.4) |
| ERM + RS | | 83.0(0.8) | 88.1(0.6) | **82.8(3.3)** | **91.2(0.5)** | **95.8(0.2)** |
| Gain | | | | +47.7(7.8) | +13.3(2.0) | +0.4(0.2) |
| CR | | - | - | 70.6(6.0) | 88.7(1.2) | 94.2(0.7) |
| CR + RS | | 82.9(0.5) | 88.2(0.3) | **82.7(5.2)** | **91.0(1.0)** | **95.8(0.5)** |
| Gain | | | | +12.2(7.5) | +2.2(1.3) | +1.6(0.4) |
| SUBY (Idrissi et al., 2022) | | - | - | 65.7(3.9) | 87.5(0.9) | 94.5(0.7) |
| SUBY + RS | | 81.5(1.0) | 87.4(0.1) | **80.8(2.9)** | **90.5(0.8)** | **95.5(0.6)** |
| Gain | | | | +15.1(3.0) | +3.0(0.9) | +1.1(0.6) |
| LfF (Nam et al., 2020) | | - | - | 55.6(6.6) | 81.5(2.8) | 92.4(0.8) |
| LfF + RS | | 74.1(3.5) | 79.7(2.6) | **78.7(4.1)** | **85.4(2.4)** | **93.4(0.7)** |
| Gain | | | | +23.2(2.5) | +4.0(0.8) | +1.0(0.2) |
| JTT (Liu et al., 2021) | | - | - | 75.1(3.6) | 85.9(1.4) | 89.8(0.8) |
| JTT + RS | | 77.3(0.7) | 81.9(0.7) | **82.9(2.3)** | **87.6(0.5)** | **90.3(1.3)** |
| Gain | | | | +7.8(3.0) | +1.7(0.4) | +0.6(0.1) |
| Group DRO (Sagawa et al., 2020) | ✓ | - | - | 88.4(2.3) | 92.0(0.4) | 93.2(0.8) |
| Group DRO + RS | ✓ | 87.3(0.2) | 88.3(0.2) | **89.7(1.2)** | **92.3(0.1)** | **93.9(0.3)** |
| Gain | | | | +1.4(1.0) | +0.4(0.2) | +0.7(0.5) |
| GR | ✓ | - | - | 88.6(1.9) | 92.0(0.4) | 92.9(0.8) |
| GR + RS | ✓ | 86.9(0.4) | 88.4(0.2) | **90.0(1.6)** | **92.4(0.5)** | **93.8(0.4)** |
| Gain | | | | +1.4(1.1) | +0.5(0.4) | +0.8(0.5) |
| SUBG (Idrissi et al., 2022) | ✓ | - | - - | 87.8(1.2) | 90.4(1.2) | 91.9(0.3) |
| SUBG + RS | ✓ | 83.6(1.6) | 87.5(0.7) | **88.3(0.7)** | **90.9(0.5)** | **93.9(0.2)** |
| Gain | | | | +0.5(0.4) | +0.5(0.5) | +1.9(0.6) |

robust coverage measures the area under the Pareto frontier of the robust-average accuracy trade-off curve, where the maximum operation in (4) finds the Pareto optimum for each threshold, as visualized in Figure 5c and 5d. Note that the robust accuracies such as WA and UA are two popular target metrics and we can adjust class-specific scaling factors depending on the objectives. Please refer to Appendix B.1 for more discussions.

## 3.4 INSTANCE-WISE ROBUST SCALING FOR IMPROVING GROUP ROBUSTNESS

We claim that the optimal scaling factor can be applied adaptively to each test example and has the potential to overcome the trade-off and improve accuracy even further. Previous approaches (Seo et al., 2022a; Sohoni et al., 2020) have shown the capability to identify hidden spurious attributes via clustering on the feature space for debiased representation learning. Likewise, we take advantage of feature clustering for adaptive robust scaling; we obtain the optimal class-specific scaling factors based on the cluster membership for each sample. The overall algorithm of instance-wise robust scaling (IRS) is described as follows.

1. Clustering the examples in the validation split on the feature space and store the centroids of the clusters.

2. Find the optimal scaling factors for each cluster in the validation split.

3. Assign the examples in the test split to the clusters by selecting their nearest centroids obtained from step 1.

4. For each sample in the test split, apply the optimal scaling factor obtained in step 2 based on its cluster membership.

In step 1, we use the naïve $k$-means algorithm for clustering. The number of clusters $K$ is a hyperparameter but it can be easily tuned by selecting the one that gives the highest robust coverage in the validation set. Empirically, a sufficient number of $K > 10$ gives stable and superior results, compared to the original robust scaling.

Table 2: Experimental results of robust scaling (RS) on the Waterbirds dataset using ResNet-50 with the average of three runs (standard deviations in parenthesis). *Gain* indicates the average (standard deviations) of performance improvement of RS for each run.

| Method | Group Supervision | Robust Coverage | | Accuracy (%) | | |
|---|---|---|---|---|---|---|
| | | Worst-group | Unbiased | Worst-group | Unbiased | Average |
| ERM | | - | - | 76.3(0.8) | 89.4(0.6) | 97.2(0.2) |
| ERM + RS | | 76.1(1.4) | 82.6(1.3) | **81.6(1.9)** | **89.8(0.5)** | **97.5(0.1)** |
| Gain | | | | +5.3(1.3) | +0.4(0.4) | +0.4(0.2) |
| CR | | - | - | 76.1(0.7) | 89.1(0.7) | 97.1(0.3) |
| CR + RS | | 73.6(2.3) | 82.0(1.5) | **79.4(2.4)** | **89.4(1.0)** | **97.5(0.3)** |
| Gain | | | | +3.4(1.8) | +0.3(0.4) | +0.4(0.1) |
| SUBY (Idrissi et al., 2022) | | - | - | 72.8(4.1) | 84.9(0.4) | 93.8(1.5) |
| SUBY + RS | | 72.5(1.0) | 81.2(1.4) | **75.9(4.4)** | **86.3(0.9)** | **95.5(0.2)** |
| Gain | | | | +3.4(1.8) | +0.3(0.4) | +1.7(1.1) |
| LfF (Nam et al., 2020) | | - | - | 77.0(2.7) | 87.1(1.9) | 93.4(1.8) |
| LfF + RS | | 75.7(2.7) | 80.9(0.4) | **79.5(2.5)** | **88.2(1.1)** | **94.8(1.9)** |
| Gain | | | | +2.6(2.1) | +1.1(0.9) | +1.4(1.2) |
| JTT (Liu et al., 2021) | | - | - | 86.7(0.3) | 90.2(0.2) | 92.6(0.3) |
| JTT + RS | | 83.0(0.5) | 84.6(0.6) | **88.2(0.7)** | **90.3(0.2)** | **92.9(0.4)** |
| Gain | | | | +1.4(0.7) | +0.1(0.1) | +0.4(0.2) |
| Group DRO (Sagawa et al., 2020) | ✓ | - | - | 88.0(1.0) | 92.5(0.9) | 95.8(1.8) |
| Group DRO + RS | | 83.4(1.1) | 87.4(1.4) | **89.1(1.7)** | **92.7(0.8)** | **96.4(1.5)** |
| Gain | | | | +1.1(0.8) | +0.2(0.1) | +0.5(0.5) |
| GR | ✓ | - | - | 86.1(1.3) | 89.3(0.9) | 95.1(1.3) |
| GR + RS | | 83.7(0.3) | 86.8(0.7) | **89.3(1.3)** | **92.0(0.7)** | **95.4(1.3)** |
| Gain | | | | +3.2(2.0) | +0.7(0.6) | +0.4(0.2) |
| SUBG (Idrissi et al., 2022) | ✓ | - | - | 86.5(0.9) | 88.2(1.2) | 87.3(1.1) |
| SUBG + RS | | 80.6(2.0) | 82.3(2.0) | **87.1(0.7)** | **88.5(1.2)** | **91.3(0.4)** |
| Gain | | | | +0.6(0.5) | +0.3(0.3) | +4.0(0.9) |

## 4 EXPERIMENTS

### 4.1 EXPERIMENTAL SETUP

**Implementation details** We adopt a ResNet-18 and ResNet-50 (He et al., 2016), which is pre-trained on ImageNet (Deng et al., 2009), as our backbone network for CelebA and Waterbirds datasets, respectively. We train our models using the stochastic gradient descent method with the Adam optimizer for 50 epoch, where the learning rate is $1 \times 10^{-4}$, the weight decay is $1 \times 10^{-4}$, and the batch size is 128. We adopt the standard K-means clustering and set the number of clusters $K = 20$ in Section 3.4 for all experiments. We select the final model which gives the best unbiased coverage in the validation split. Our algorithms are implemented in the Pytorch (Paszke et al., 2019) framework and all experiments are conducted on a single unit of NVIDIA Titan XP GPU. Following previous works (Sagawa et al., 2020; Liu et al., 2021), we report the adjusted average accuracy as the average accuracy for Waterbirds dataset; we first calculate the accuracy for each group and then report the weighted average, where the weights corresponds to the relative portion of each group in training set. Please refer to the supplementary file for the dataset usage.

**Evaluation metrics** Following prior works, we evaluate all the compared algorithms with the average, unbiased and worst-group accuracies, and also with the proposed unbiased and worst-group coverages for a comprehensive view.

### 4.2 RESULTS

**Plugging RS into the existing methods** Table 1 and 2 present the experimental results of our original robust scaling (RS) on top of the existing approaches (CR, SUBY, LfF, JTT, Group DRO, GR, SUBG)[2] on the CelebA and Waterbirds datasets, respectively. In these tables, RS is applied to maximize each target metric (worst-group, unbiased, and average accuracies) independently.[3] We conducted three experimental runs and reported the average and the standard deviation of the results.

---

[2]Please refer to Appendix A for a brief introduction of comparisons.

[3]Note that, since our robust scaling strategy is a simple post-processing technique, we do not need to retrain the model for each target metric and the cost is negligible, *e.g.*, only a few seconds for each target metric.

Table 3: Results of our robust scaling methods on the CelebA and Waterbirds datasets with the average of three runs (standard deviations in parenthesis). Blue color denotes the target metric that the robust scaling aims to maximize. Compared to RS, IRS improves the overall trade-off.

| Dataset | Method | Robust Coverage | | Accuracy (%) | | | Accuracy (%) | | |
|---|---|---|---|---|---|---|---|---|---|
| | | Worst. | Unbiased | Worst. | Unbiased | Average | Worst. | Unbiased | Average |
| CelebA | ERM | - | - | 34.5(6.1) | 77.7(1.8) | **95.5(0.4)** | 34.5(6.1) | 77.7(1.8) | 95.5(0.4) |
| | ERM + RS | 83.0(0.7) | 88.1(0.5) | 82.1(3.7) | 91.1(0.6) | 92.2(1.3) | **45.0(7.4)** | **81.7(1.8)** | **95.8(0.2)** |
| | ERM + IRS | **83.4(0.1)** | **88.4(0.4)** | **87.2(2.0)** | **91.7(0.2)** | 91.5(0.8) | 44.1(4.2) | 81.3(0.8) | **95.8(0.1)** |
| | CR | - | - | 70.6(6.0) | 88.7(1.2) | **94.2(0.7)** | **70.6(6.0)** | **88.7(1.2)** | 94.2(0.7) |
| | CR + RS | 82.9(0.5) | 88.2(0.3) | 82.7(5.2) | 91.0(1.0) | 91.7(1.3) | 48.5(8.9) | 82.5(2.2) | **95.8(0.1)** |
| | CR + IRS | **83.6(1.1)** | **88.6(0.5)** | **84.8(1.5)** | **91.3(0.4)** | 90.7(1.3) | 48.8(9.1) | 82.7(2.4) | **95.8(0.1)** |
| | GroupDRO | - | - | 88.4(2.3) | 92.0(0.4) | 93.2(0.8) | **88.4(2.3)** | **92.0(0.4)** | 93.2(0.8) |
| | GroupDRO + RS | 87.3(0.2) | 88.3(0.2) | 89.7(1.2) | 92.3(0.1) | **93.7(0.5)** | 64.9(3.3) | 85.1(0.7) | 93.9(0.3) |
| | GroupDRO + IRS | **87.5(0.4)** | **88.4(0.2)** | **90.0(2.3)** | **92.6(0.6)** | 93.5(0.4) | 60.4(5.4) | 84.4(0.6) | **94.7(0.3)** |
| Waterbirds | ERM | - | - | 76.3(0.8) | 89.4(0.6) | **97.2(0.2)** | 76.3(0.8) | 89.4(0.6) | 97.2(0.2) |
| | ERM + RS | 76.1(0.4) | 82.6(0.3) | 81.6(1.9) | 89.8(0.5) | **97.2(0.2)** | **79.1(2.7)** | **89.7(0.6)** | 97.5(0.1) |
| | ERM + IRS | **83.4(1.1)** | **86.9(0.4)** | **89.3(0.5)** | **92.7(0.4)** | 94.1(0.3) | 77.6(7.0) | 89.6(1.1) | **97.5(0.3)** |
| | CR | - | - | 76.1(0.7) | 89.1(0.7) | **97.1(0.5)** | 76.1(0.7) | 89.1(0.7) | 97.1(0.3) |
| | CR + RS | 73.6(2.3) | 82.0(1.5) | 79.4(2.4) | 89.4(1.0) | 96.8(0.3) | 76.4(1.5) | **89.3(0.8)** | **97.5(0.3)** |
| | CR + IRS | **84.2(2.5)** | **88.3(1.0)** | **88.2(2.7)** | **92.1(0.7)** | 95.7(1.1) | **77.3(4.7)** | 88.6(1.2) | 97.4(0.2) |
| | GroupDRO | - | - | 88.0(1.0) | 92.5(0.9) | **95.8(1.8)** | **88.0(1.0)** | **92.5(0.9)** | 95.8(1.8) |
| | GroupDRO + RS | 83.4(1.1) | 87.4(1.4) | 89.1(1.7) | 92.7(0.8) | **96.4(1.5)** | 80.9(4.4) | 91.3(1.0) | **97.1(0.3)** |
| | GroupDRO + IRS | **86.3(2.3)** | **90.1(2.6)** | **90.8(1.3)** | **93.9(0.2)** | 96.0(0.6) | 83.2(1.7) | 91.5(0.8) | **97.1(0.4)** |

Group supervision indicates that the method requires training examples with group supervision. As shown in the tables, no matter what the backbone method is, our robust scaling strategy consistently improves the performance for all target metrics. Based on the robust coverage and robust accuracy after scaling, LfF and JTT are not superior to ERM on the CelebA dataset, though their initial robust accuracies without scaling are much higher than ERM. GR and SUBG are reweighting and subsampling baselines based on group frequency, respectively. Although two method achieve competitive robust accuracy on the Waterbirds dataset, the average accuracy of SUBG is far below than GR, mainly because SUBG drops a large portion of training samples to make all groups have the same size. This supports our argument that we need to evaluate in a more comprehensive way beyond robust accuracy. On the other hand, the methods that leverages group supervision (Group DRO, GR) achieve better robust coverage results than others on both datasets, which validates that group supervision helps to improve the trade-off itself. For the group supervised methods, our scaling provides relatively small performance gains in robust accuracy, as the gaps between robust and average accuracies are already small and the original results are already close to the optimal for maximizing robust accuracy.

**Instance-wise Robust Scaling** We evaluate all the proposed methods on the CelebA and Waterbirds datasets and present the results in Table 3. We employ ERM, a class reweighting scheme (CR), and Group DRO as baselines, and apply our robust scaling methods, including the standard robust scaling (RS) and instance-wise robust scaling (IRS), to them. We test our robust scaling strategies with two scenarios, each aimed at maximizing unbiased or average accuracies, respectively, where each target metric is marked in blue in the tables. Compared to RS, IRS improve the robust coverage itself as well as the accuracy of the target metric for each scenario. In particular, although Group DRO have already achieved high robust accuracy and coverage, IRS further improves their results on both datasets. This validates that our instance-wise class-specific scaling strategy is effective to overcome the trade-off between robust and average accuracies and truly improve the group robustness of baseline models.

## 4.3 ANALYSIS

**Visualization of the trade-off curve and its Pareto frontier** We visualize the robust-average accuracy trade-off curve and its corresponding robust coverage curve in Fig. 5 on the CelebA dataset. In Fig. 5a and 5b, the black markers indicate the points where the scaling factor is not applied, indicating that there is room for improvement in robust accuracy along the trade-off curve. Fig. 5c and 5d present the corresponding robust coverage curves, which represents the Pareto frontiers of

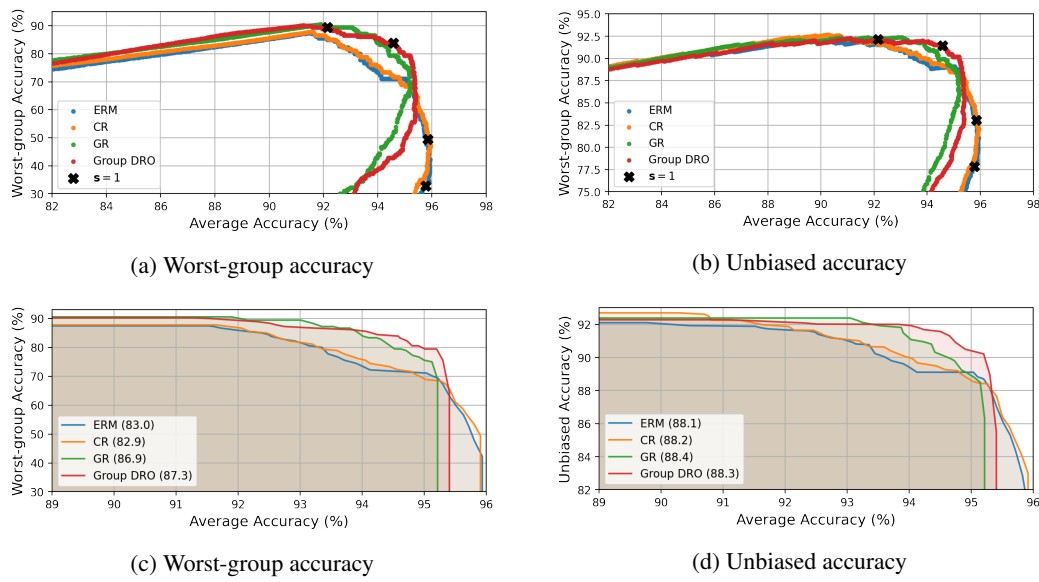

(a) Worst-group accuracy

(b) Unbiased accuracy

(c) Worst-group accuracy

(d) Unbiased accuracy

Figure 5: The robust-average accuracy trade-off curves ((a), (b)) and the corresponding robust coverage curves ((c), (d)), respectively, on the CelebA dataset. The curves in (c) and (d) represent the Pareto frontiers of the curves in (a) and (b), respectively. In (c) and (d), the numbers in the legend denote the robust coverage, which measures the area under the curve.

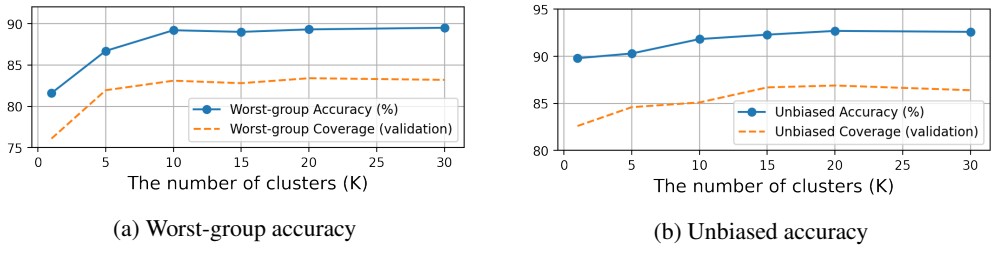

(a) Worst-group accuracy

(b) Unbiased accuracy

Figure 6: Sensitivity analysis on the number of clusters in the instance-wise robust scaling on the Waterbirds dataset. The robust coverage in the validation split (orange line) tends to be similar to the robust accuracy in the test split (blue line).

Fig. 5a and 5b, respecitvely. The area under the curve in Fig. 5c and 5d indicates the robust coverage, which is reported in the legend.

**Sensitivity analysis on the number of clusters** We conduct the ablation study on the number of clusters for feature clustering in our instance-wise robust scaling method on the Waterbirds dataset. Fig. 6 shows that the worst-group and unbiased accuracies are stable with a sufficient number of $K > 10$. We also plot the robust coverage results in the validation split, which tends to almost consistent with the robust accuracy. This presents that we can select the optimal $K$ based on the robust coverage results of the validation split.

## 5 CONCLUSION

We presented a simple but effective post-processing framework that provides a novel perspective of group robustness. Our work started from the observation that there exists a clear trade-off between robust and average accuracies in existing works, and argued that comparing only the robust accuracy should be regarded as incomplete. To deal with this issue, we first proposed the robust scaling strategy, which can capture the full landscape of trade-off and identify the desired performance point on the trade-off curve. Based on this strategy, we introduced a novel convenient measurement

that summarizes the trade-off from a Pareto optimal perspective for the comprehensive evaluation of group robustness. Moreover, we proposed an instance-wise robust scaling algorithm with adaptive scaling factors, which is effective to improve the trade-off itself. We analyzed the characteristics of existing methods and demonstrated the effectiveness of our proposed frameworks and measurements with extensive experiments. We believe that the proposed approaches are helpful for analyzing the exact behavior of existing debiasing techniques and paving the way in the future research direction.

**Ethics Statement**   As model learning focuses unconditionally on improving the overall performance, there is room for risks such as fairness or bias issues caused by dataset or algorithmic bias. With the careful and comprehensive consideration of robust performance such as worst-group or unbiased accuracies, we believe our framework can help to alleviate this potentially harmful risks in model use. We also acknowledge that we have read and commit to adhering to the ICLR code of ethics.

**Reproducibility Statement**   We provide the dataset usage, implementation details, and evaluation metric in Section 4.1. We reported all experimental results in the main tables based on the average of three runs with standard deviation.

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

## A  COMPARISONS

Below is a brief introduction of the comparisons used in our experiments.

**ERM**   Given a loss function $\ell(\cdot)$, the objective of empirical risk minimization is optimizing the following loss over training data:

$$\min_\theta \Big\{ \frac{1}{n} \sum_{i=1}^n \ell(f_\theta(x_i), y_i) \Big\}. \tag{5}$$

**Class reweighting (CR)**   To mitigate the class imbalance issue, we can simply reweight the samples based on the inverse of class frequency in the training split,

$$\min_\theta \Big\{ \frac{1}{n} \sum_{i=1}^n \omega_i \ell(f_\theta(x_i), y_i) \Big\} \text{ where } \omega_i = \frac{n}{\sum_j \mathbb{1}(y_j = y_i)}. \tag{6}$$

**LfF**   Motivated by the observation that bias-aligned samples are more easily learned, LfF (Nam et al., 2020) simultaneously trains a pair of neural network $(f_B, f_D)$. The biased model $f_B$ is trained with generalized cross-entropy loss which intends to amplify bias, while the debiased model $f_D$ is trained with a standard cross-entropy loss, where each sample $(x_i, y_i)$ is reweighted by the following relative difficulty score:

$$\omega_i = \frac{\ell(f_\theta^B(x_i), y_i)}{\ell(f_\theta^B(x_i), y_i) + \ell(f_\theta^D(x_i), y_i)}. \tag{7}$$

**JTT**   JTT (Liu et al., 2021) consists of two-stage procedures. In the first stage, JTT trains a standard ERM model $\hat{f}(\cdot)$ for several epochs and identifies an error set $E$ of training examples that are misclassified:

$$E := \{(x_i, y_i) \text{ s.t. } \hat{f}(x_i) \neq y_i\}. \tag{8}$$

Next, they train a final model $f_\theta(\cdot)$ by upweighting the examples in the error set $E$ as

$$\min_\theta \Big\{ \lambda_{\text{up}} \sum_{(x,y) \in E} \ell(f_\theta(x), y) + \sum_{(x,y) \notin E} \ell(f_\theta(x), y) \Big\}. \tag{9}$$

**Group DRO**   Group DRO (Sagawa et al., 2020) aims to minimize the empirical worst-group loss formulated as:

$$\min_\theta \Big\{ \max_{g \in \mathcal{G}} \frac{1}{n_g} \sum_{i | g_i = g}^{n_g} \ell(f_\theta(x_i), y_i) \Big\} \tag{10}$$

where $n_g$ is the number of samples assigned to $g^{\text{th}}$ group. Unlike previous approaches, group DRO requires group annotations $g = (y, a)$ on the training split.

**Group reweighting (GR)**   Using group annotations, we can extend class reweighting method to group reweighting one based on the inverse of group frequency in the training split, *i.e.*,

$$\min_\theta \Big\{ \frac{1}{n} \sum_{i=1}^n \omega_i \ell(f_\theta(x_i), y_i) \Big\} \text{ where } \omega_i = \frac{n}{\sum_j \mathbb{1}(y_j = y_i, a_j = a_i)} \tag{11}$$

**SUBY/SUBG**   To mitigate the data imbalance issue, SUBY subsample majority classes, so all classes have the same size with the smallest class on the training dataset, as in Idrissi et al. (2022). Similarly, SUBG subsample majority groups.

Table A4: Realized robust coverage results on the Waterbirds and CelebA datasets with the average of three runs (standard deviations in parenthesis).

| Dataset | Method | Robust Coverage | | Realized Robust Coverage | |
|---|---|---|---|---|---|
| | | Worst-group | Unbiased | Worst-group | Unbiased |
| Waterbirds | ERM | 70.3(1.3) | 79.4(0.7) | 69.0(1.5) | 78.7(0.8) |
| Waterbirds | CR | 68.9(1.1) | 78.5(0.5) | 67.8(1.2) | 77.9(0.4) |
| Waterbirds | Group DRO (Sagawa et al., 2020) | 80.8(0.6) | 85.2(0.1) | 78.6(1.0) | 83.8(0.4) |
| Waterbirds | GR | 78.8(5.6) | 83.7(0.7) | 77.9(1.4) | 82.8(0.8) |
| CelebA | ERM | 78.9(1.7) | 86.0(0.6) | 75.9(2.2) | 85.4(0.7) |
| CelebA | CR | 77.2(2.8) | 85.6(0.9) | 71.8(1.3) | 85.0(0.6) |
| CelebA | Group DRO (Sagawa et al., 2020) | 84.2(0.6) | 86.7(0.5) | 81.0(1.7) | 86.1(0.2) |
| CelebA | GR | 84.2(0.5) | 87.5(0.3) | 81.2(1.6) | 87.0(0.5) |

## B  CLARIFICATION

### B.1  ROBUST COVERAGE IN THE TEST SPLIT

In Sec. 4.3 of the main paper, we defined robust coverage as

$$\text{Coverage} := \int_{c=0}^{1} \max_{\mathbf{s}} \big\{ \text{WA}^{\mathbf{s}} | \text{AA}^{\mathbf{s}} \geq c \big\} dc \approx \sum_{d=0}^{D-1} \frac{1}{D} \max_{\mathbf{s}} \big\{ \text{WA}^{\mathbf{s}} | \text{AA}^{\mathbf{s}} \geq \frac{d}{D} \big\}. \quad (12)$$

The robust coverage can be directly calculated in the validation split, but unfortunately, it is basically unavailable in the test split. This is because we need to know the values of $\text{WA}^{\mathbf{s}}$ in advance to conduct max operation in (12), but we cannot use the information in the test split.

To bypass this issue, we report the robust coverage of validation split in this paper, which tends to be similar to those of the test split. We also validate the reliability of the robust coverage of the test split. We first obtain a set of optimal scaling factors for each threshold in the validation split $\mathcal{S}_{\text{val}}$ as

$$\mathcal{S}_{\text{val}} := \Big\{ \max_{\mathbf{s}} \big\{ \text{WA}_{\text{val}}^{\mathbf{s}} | \text{AA}_{\text{val}}^{\mathbf{s}} \geq \frac{d}{D} \big\} \ \text{for} \ 0 \leq d \leq D-1 \Big\}, \quad (13)$$

then the realized robust coverage of test split is calculated by

$$\text{Realized Coverage} := \frac{1}{|\mathcal{S}_{\text{val}}|} \sum_{\mathbf{s} \in \mathcal{S}_{\text{val}}} \text{WA}_{\text{test}}^{\mathbf{s}}. \quad (14)$$

Table A4 presents the original robust coverage and realized robust coverage results on the test splits of Waterbirds and CelebA datasets. Both coverage results are almost similar, because the optimal scaling factors identified in the validation split are close to optimal in the test split as well.

## C  ATTRIBUTE-SPECIFIC ROBUST SCALING WITH GROUP SUPERVISION

If the supervision of group (spurious-attribute) information can be utilized during our robust scaling, it will provide flexibility to further improve the performance. To this end, we first partition the examples based on the values of spurious attributes and find the optimal scaling factors for each partition separately. Like as the original robust scaling procedure, we obtain the optimal scaling factors for each partition in the validation split and apply them to the test split. However, this partition-wise scaling is basically unavailable because we do not know the spurious attribute values of the examples in the test split and thus cannot partition them, In other words, we need to estimate the spurious-attribute values in the test split for partitioning. To conduct attribute-specific robust scaling (ARS), we follow a simple algorithm described below:

1. Partition the examples in the validation split by the values of the spurious attribute.

2. Find the optimal scaling factors for each partition in the validation split.

3. Train an independent estimator model to classify spurious attribute.

Table A5: Results of the attribute-specific robust scaling (ARS) on the CelebA and Waterbirds datasets with the average of three runs (standard deviations in parenthesis), where ARS is applied to maximize each target metric independently. Note that our post-processing strategy, ARS, allows ERM to achieve competitive performance to Group DRO that utilizes the group supervision during training.

| | | Robust Coverage | | Accuracy (%) | | |
|---|---|---|---|---|---|---|
| Dataset | Method | Worst. | Unbiased | Worst. | Unbiased | Average |
| | ERM | - | - | 34.5(6.1) | 77.7(1.8) | 95.5(0.4) |
| CelebA | ERM + ARS | **87.6(1.0)** | **89.0(0.2)** | **88.5(1.8)** | 91.9(0.3) | **95.8(0.1)** |
| | Group DRO | 87.3(0.2) | 88.3(0.2) | 88.4(2.3) | **92.0(0.4)** | 93.2(0.8) |
| | ERM | - | - | 76.3(0.8) | 89.4(0.6) | 97.2(0.2) |
| Waterbirds | ERM + ARS | **84.4(1.9)** | **87.8(1.7)** | **89.3(0.4)** | **92.5(0.4)** | **97.5(1.0)** |
| | Group DRO | 83.4(1.1) | 87.4(2.3) | 88.0(1.0) | **92.5(0.9)** | 95.8(1.8) |

Table A6: Effects of the size of group-labeled examples on the attribute-specific robust scaling on the CelebA dataset. Group-labeled size denotes a ratio of group-labeled samples among all training examples for training estimators. Spurious accuracy indicates the average accuracy of spurious-attribute classification using the estimators on the test split.

| | Accuracy (%) | Accuracy (%) | | | Robust Coverage | |
|---|---|---|---|---|---|---|
| Group-labeled size | Spurious | Worst-group | Unbiased | Average | Worst-group | Unbiased |
| 100% | 98.4 | 89.1(3.0) | 92.4(1.1) | 93.1(1.2) | 87.6(1.0) | 89.0(0.5) |
| 10% | 97.7 | 88.5(1.8) | 91.9(0.3) | 92.8(0.6) | 86.8(0.4) | 89.0(0.2) |
| 1% | 95.8 | 88.5(1.8) | 91.9(0.3) | 92.9(0.6) | 87.1(0.3) | 89.0(0.2) |
| 0.1% | 92.6 | 88.4(2.1) | 91.8(0.5) | 92.4(0.8) | 87.1(0.3) | 89.0(0.2) |

4. Estimate the spurious attribute values of the examples in the test split using the estimator, and partition the test samples according to their estimated spurious attribute values.

5. For each sample in the test split, apply the optimal scaling factors obtained in step 2 based on its partition.

To find a set of scale factors corresponding to each partition, we adopt a naïve greedy algorithm that performed in one partition at a time. This attribute-specific robust scaling further increases the robust accuracy compared to the original robust scaling, and also improves the robust coverage, as shown in Table A5. Note that our attribute-specific scaling strategy allows ERM to match the supervised state-of-the-art approach, Group DRO (Sagawa et al., 2020).

One limitation is that it requires the supervision of spurious attribute information to train the estimator model in step 3. However, we notice that only a very few examples with the supervision is enough to train the spurious-attribute estimator, because it is much easier to learn as the word "spurious correlation" suggests. To determine how much the group-labeled data is needed, we train several spurious-attribute estimators by varying the number of group-labeled examples, and conduct ARS using the estimators. Table A6 validates that, compared to the overall training dataset size, a very small amount of group-labeled examples is enough to achieve high robust accuracy.

## D    EXPERIMENTAL DETAILS

### D.1    DATASETS

CelebA (Liu et al., 2015) is a large-scale dataset for face image recognition, consisting of 202,599 celebrity images, with 40 attributes labeled on each image. Among the attributes, we primarily examine *hair color* and *gender* attributes as a target and spurious attributes, respectively. We follow the original train-validation-test split (Liu et al., 2015) for all experiments in the paper. Waterbirds (Sagawa et al., 2020) is a synthesized dataset, which are created by combining bird images in the CUB dataset (Wah et al., 2011) and background images from the Places dataset (Zhou et al., 2017), consisting of 4,795 training examples. The two attributes—one is the type of bird, {waterbird,

Table A7: Ablation study on the size of validation set in our robust scaling strategies on the CelebA dataset.

| Method | Valid set size | Worst-group Acc. | Unbiased Acc. | Average Acc. |
|--------|----------------|------------------|---------------|--------------|
| ERM    | -              | 34.5(6.1)        | 77.7(1.8)     | 95.5(0.4)    |
| + RS   | 100%           | 82.8(3.3)        | 91.2(0.5)     | 95.8(0.2)    |
| + RS   | 50%            | 83.3(3.7)        | 91.5(0.9)     | 95.8(0.2)    |
| + RS   | 10%            | 82.4(4.3)        | 91.4(0.8)     | 95.8(0.2)    |
| + RS   | 1%             | 79.2(10.3)       | 90.8(2.2)     | 95.5(0.4)    |
| + IRS  | 100%           | 88.7(0.9)        | 92.0(0.3)     | 95.8(0.1)    |
| + IRS  | 50%            | 86.9(2.0)        | 91.8(0.4)     | 95.9(0.2)    |
| + IRS  | 10%            | 84.4(6.3)        | 91.4(1.0)     | 95.6(0.4)    |
| + IRS  | 1%             | 60.4(14.4)       | 85.8(3.2)     | 94.7(1.5)    |

Table A8: Variations of robust scaling methods tested on the FairFace dataset.

| Method | Cost | Worst-group Acc. | Unbiased Acc. | Average Acc. |
|--------|------|------------------|---------------|--------------|
| ERM | − | 15.8 | 47.0 | 54.1 |
| + RS (2 super classes) | $\mathcal{O}(n)$ | 18.6 | 51.8 | 52.9 |
| + RS (greedy search) | $\mathcal{O}(n)$ | **19.2** | 52.3 | **53.3** |
| + RS (full grid search) | $\mathcal{O}(n^9)$ | 19.0 | **52.8** | 53.1 |

landbird} and the other is background places, {water, land}, are used for the experiments with this dataset.

## D.2 CLASS-SPECIFIC SCALING

To identify the optimal points, we obtain a set of the average and robust accuracy pairs using a wide range of the class-specific scaling factors, *i.e.*, $\mathbf{s}_i = (1.05)^n$ for $-200 \leq n \leq 200$ for $i^{\text{th}}$ class. Note that we search for the scaling factor of each class in a greedy manner, as stated in Section 3.2.

## D.3 HYPERPARAMETER TUNING

We tune the learning rate in $\{10^{-3}, 10^{-4}, 10^{-5}\}$ and the weight decay in $\{1.0, 0.1, 10^{-2}, 10^{-4}\}$ for all baselines. We used 0.5 of $q$ for LfF. For JTT, we searched $\lambda_{\text{up}}$ in $\{20, 50, 100\}$ and updated the error set every epoch for CelebA dataset and every 60 epochs for Waterbirds dataset. For Group DRO, we tuned $C$ in $\{0, 1, 2, 3, 4\}$, and used 0.1 of $\eta$.

## E ADDITIONAL ANALYSIS

**Ablative study on the size of validation set** Our robust scaling strategies need a held-out validation set to identify the optimal scaling factors, like as other existing approaches for early stopping and hyperparameter tuning. Note that early stopping is essential to achieve high robust accuracy in other approaches. To validate the robustness of our frameworks, we conduct the ablation study by varying the size of validation set. Table A7 presents the ablative results using ERM baseline on the CelebA dataset, where the size of validation set is varied by $\{100\%, 50\%, 10\%, 1\%\}$. As shown in the table, with only 10% or 50% of the validation set, both our robust scaling (RS) and instance-wise robust scaling (IRS) achieves almost competitive performance with the results using the full size of validation set. Surprisingly, even only 1% of the validation set is enough for RS to achieve sufficiently high robust accuracy, but inevitably entails a large variance of results. On the other hand, IRS suffers from performance degradation when only 1% of the validation set is available. This is mainly because IRS takes advantage of feature clustering on the validation set, which needs some validation samples. Nevertheless, our robust scaling strategies can achieve meaningful performance improvement with a limited number of validation samples for all cases, which validates the robustness of our method.

Table A9: Results of our robust scaling methods on top of various baselines on the CelebA and Waterbirds datasets with the average of three runs (standard deviations in parenthesis). Blue color denotes the target metric that the robust scaling aims to maximize. Compared to RS, IRS improves the overall trade-off.

| Dataset | Method | Robust Coverage | | Accuracy (%) | | | Accuracy (%) | | |
|---|---|---|---|---|---|---|---|---|---|
| | | Worst. | Unbiased | Worst. | Unbiased | Average | Worst. | Unbiased | Average |
| CelebA | ERM | - | - | 34.5(6.1) | 77.7(1.8) | **95.5(0.4)** | 34.5(6.1) | 77.7(1.8) | 95.5(0.4) |
| | ERM + RS | 83.0(0.7) | 88.1(0.5) | 82.1(3.7) | 91.1(0.6) | 92.2(1.3) | **45.0(7.4)** | **81.7(1.8)** | **95.8(0.2)** |
| | ERM + IRS | **83.4(0.1)** | **88.4(0.4)** | **87.2(2.0)** | **91.7(0.2)** | 91.5(0.8) | 44.1(4.2) | 81.3(0.8) | **95.8(0.1)** |
| | CR | - | - | 70.6(6.0) | 88.7(1.2) | **94.2(0.7)** | 70.6(6.0) | 88.7(1.2) | 94.2(0.7) |
| | CR + RS | 82.9(0.5) | 88.2(0.3) | 82.7(5.2) | 91.0(1.0) | 91.7(1.3) | 48.5(8.9) | 82.5(2.2) | **95.8(0.1)** |
| | CR + IRS | **83.6(1.1)** | **88.6(0.5)** | **84.8(1.5)** | **91.3(0.4)** | 90.7(1.3) | 48.8(9.1) | 82.7(2.4) | **95.8(0.1)** |
| | SUBY | - | - | 65.7(3.9) | 87.5(0.9) | **94.5(0.7)** | 65.7(3.9) | 87.5(0.9) | 94.5(0.7) |
| | SUBY + RS | 81.5(1.0) | 87.4(0.1) | 80.8(2.9) | 90.5(0.8) | 91.1(1.7) | 45.4(6.7) | 81.4(2.0) | **95.5(0.0)** |
| | SUBY + IRS | **82.3(1.1)** | **87.8(0.2)** | 82.3(2.0) | **90.8(0.8)** | 90.7(1.9) | 46.0(6.9) | 81.5(2.1) | **95.5(0.1)** |
| | SUBG | - | - | 87.8(1.2) | 90.4(1.2) | **91.9(0.3)** | 87.8(1.2) | 90.4(1.2) | 91.9(0.3) |
| | SUBG + RS | 83.6(1.6) | 87.5(0.7) | 88.3(0.7) | 90.9(0.5) | 90.6(1.0) | 67.8(6.5) | 85.2(2.0) | 93.9(0.2) |
| | SUBG + IRS | **84.5(0.8)** | **87.9(0.1)** | **88.7(0.6)** | **91.0(0.3)** | 90.6(0.8) | 68.5(6.5) | 85.5(1.9) | **94.0(0.2)** |
| | GR | - | - | 88.6(1.9) | 92.0(0.4) | **92.9(0.8)** | 88.6(1.9) | 92.0(0.4) | 92.9(0.8) |
| | GR + RS | 86.9(0.4) | 88.4(0.2) | **90.0(1.6)** | 92.4(0.5) | 92.5(0.5) | 66.5(0.3) | 85.4(0.4) | 93.8(0.4) |
| | GR + IRS | **87.0(0.2)** | **88.6(0.2)** | 90.0(2.3) | **92.6(0.6)** | 92.5(0.4) | 62.0(5.3) | 84.5(0.7) | **94.2(0.3)** |
| | GroupDRO | - | - | 88.4(2.3) | 92.0(0.4) | **93.2(0.8)** | 88.4(2.3) | 92.0(0.4) | 93.2(0.8) |
| | GroupDRO + RS | 87.3(0.2) | 88.3(0.2) | 89.7(1.2) | 92.3(0.1) | **93.7(0.5)** | 64.9(3.3) | 85.1(0.7) | 93.9(0.3) |
| | GroupDRO + IRS | **87.5(0.4)** | **88.4(0.2)** | 90.0(2.3) | **92.6(0.6)** | 93.5(0.4) | 60.4(5.4) | 84.4(0.6) | **94.7(0.3)** |
| Waterbirds | ERM | - | - | 76.3(0.8) | 89.4(0.6) | **97.2(0.2)** | 76.3(0.8) | 89.4(0.6) | 97.2(0.2) |
| | ERM + RS | 76.1(0.4) | 82.6(0.3) | 81.6(1.9) | 89.8(0.5) | **97.2(0.2)** | **79.1(2.7)** | **89.7(0.6)** | 97.5(0.1) |
| | ERM + IRS | **83.4(1.1)** | **86.9(0.4)** | **89.3(0.5)** | **92.7(0.4)** | 94.1(0.3) | 77.6(7.0) | 89.6(1.1) | **97.5(0.3)** |
| | CR | - | - | 76.1(0.7) | 89.1(0.7) | **97.1(0.5)** | 76.1(0.7) | 89.1(0.7) | 97.1(0.3) |
| | CR + RS | 73.6(2.3) | 82.0(1.5) | 79.4(2.4) | 89.4(1.0) | 96.8(0.8) | 76.4(1.5) | **89.3(0.8)** | **97.5(0.3)** |
| | CR + IRS | **84.2(2.5)** | **88.3(1.0)** | **88.2(2.7)** | **92.1(0.7)** | 95.7(1.1) | **77.3(4.7)** | 88.6(1.2) | 97.4(0.2) |
| | SUBY | - | - | 72.8(4.1) | 84.9(0.4) | **93.8(1.5)** | 72.8(4.1) | 84.9(0.4) | 93.8(1.5) |
| | SUBY + RS | 72.5(1.0) | 81.2(1.4) | 75.9(4.4) | 86.3(0.9) | **95.2(1.4)** | 70.7(5.8) | 85.4(1.6) | 95.5(0.2) |
| | SUBY + IRS | **78.8(2.7)** | **85.9(1.0)** | **82.1(4.0)** | **89.1(0.9)** | 92.6(2.2) | **74.1(4.1)** | **86.3(0.9)** | **96.2(0.6)** |
| | SUBG | - | - | 86.5(0.9) | 88.2(1.2) | **87.3(1.1)** | 86.5(0.9) | 88.2(1.2) | 87.3(1.1) |
| | SUBG + RS | 80.6(2.0) | 82.3(2.0) | 87.1(0.7) | **88.5(1.2)** | **87.9(1.1)** | 74.0(5.6) | 85.9(2.8) | 91.3(0.4) |
| | SUBG + IRS | **82.2(0.8)** | **84.1(0.8)** | **87.3(1.3)** | 88.2(1.2) | 87.6(1.2) | 70.2(1.6) | 84.5(1.0) | **93.5(0.4)** |
| | GR | - | - | 86.1(1.3) | 89.3(0.9) | **95.1(1.3)** | 86.1(1.3) | 89.3(0.9) | 95.1(1.3) |
| | GR + RS | 83.7(0.3) | 86.8(0.7) | **89.3(1.3)** | 92.0(0.7) | 93.1(3.2) | 82.2(1.3) | **90.8(0.5)** | 95.4(1.3) |
| | GR + IRS | **84.8(1.7)** | **87.4(0.4)** | 89.1(0.8) | **92.2(1.0)** | 92.9(2.1) | 82.1(1.4) | 90.5(0.7) | **95.6(0.8)** |
| | GroupDRO | - | - | 88.0(1.0) | 92.5(0.9) | **95.8(1.8)** | 88.0(1.0) | 92.5(0.9) | 95.8(1.8) |
| | GroupDRO + RS | 83.4(1.1) | 87.4(1.4) | 89.1(1.7) | 92.7(0.8) | **96.4(1.5)** | 80.9(4.4) | 91.3(1.0) | **97.1(0.3)** |
| | GroupDRO + IRS | **86.3(2.3)** | **90.1(2.6)** | **90.8(1.3)** | **93.9(0.2)** | 96.0(0.6) | 83.2(1.7) | 91.5(0.8) | **97.1(0.4)** |

**Scalability** As mentioned in Sec. 3.2, we actually search for the scaling factor of each class in a greedy manner. Hence, the time complexity increases linearly with respect to the number of classes instead of the exponential growth with the full grid search; even with 1000 classes, the whole process takes less than a few minutes in practice, which is negligible compared to the model training time. Moreover, we can reduce the computational cost even further by introducing the superclass concept and allocating a single scaling factor for each superclass. We compare three different options—greedy search, superclass-level search, and full grid search—on the FairFace dataset Kärkkäinen & Joo (2021) with 9 classes. Table A8 shows that our greedy search is as competitive as the full grid search despite the time complexity gap in several orders of magnitude in computational complexity and the superclass-level search is also effective to reduce cost. Note that the superclasses are identified by the feature similarity of class signatures.

# F ADDITIONAL RESULTS

Table A9 presents the full experimental results with our robust scaling strategies (RS, IRS) on top of existing models, which supplement Table 3 of the main paper.

