# OpenReview forum: "Revisiting Group Robustness: Class-specific Scaling is All You Need"
_ICLR.cc/2023/Conference — Submitted to ICLR 2023_

### Official Review · Reviewer_TDJz · 2022-10-20

**Confidence:** 4
**Correctness:** 2
**Technical Novelty And Significance:** 1
**Empirical Novelty And Significance:** 2
**Recommendation:** 3

**Clarity, Quality, Novelty And Reproducibility:**

See above. Reproducibility concerns are particularly around how hyperparameter searches/tuning were conducted, but the final values are reported.

**Strength And Weaknesses:**


## Major comments

* The paper lacks any motivation for the proposed method. While of course robustness is a worthy goal, why *rescaling* would be a useful or appropriate technique to advance the community toward this goal is not obvious -- and there is no discussion in the paper for why either.

* The method is not well-described. The equation (3) does not fully define the problem being solved. For example, what is really meany be a "scaling coefficient vector" -- does it need to have unit norm? Why do we take arg max in (3)? What are the practical implications of using this method? There is no discussion of this in the paper; lacking any theoretical results, there should be strong empirical or intuitive support for why this is a reasonable postprocessing choice.

* The implementations of "baseline" or prior works are not faithful, and report much lower numbers than those works on the same datasets. For example, JTT reports much higher worst-group and average accuracies on Waterbirds (Table9 listd their best as 88.6% and 93.1% in Table 9, see also Table 7,8; these are lower than in the current paper Table 2) and Group DRO reports mich higher Waterbirds worst-group and average accuracies in Table 3. I am concerned that this is due to poor hyperparameter tuning or model architecture selection; the approach to tuning is not described at all in the paper so it is difficult to tell (if no tuning were performed, this would also be a concern).

* The empirical results are weak. For example, only 4/18 comparisons in Table 2 show an improvement greater than two standard deviations, and only one of the eight comparisons on Table 3 (for the blue "target metric that robust scaling aims to maximize" columns) is greater than two standard deviations.

* Small contribution - beyond the rescaling method, the proposed metric is not a standalone contribution. I think the metric is lacking a thorough discussion or motivation.

# Minor comments

* Figure 1 needs more detail. What dataset/task is this? Are the different points different initializations, hyperparameter configurations? There is very weak evidence of a tradeoff here, but the details are far too murky to draw much of a conclusion; if there were not lines drawn on the graphs I would not conclude there was any relationship.

* Figure 2 also doesn't describe (a) the dataset/task, or (b) what models are used for each method, how they were selected, or give any notion of variability (i.e. Clopper-Pearson confidence intervals). Again, it is hard to learn much without this information.

* It seems Figure 3 is missing any reference to a baseline; can this comparison be made e.g. to ERM? Also, how are we supposed to conclude that any other scaling (s \neq 1) is an improvement based on this plot?

* Other hyperparameters also seem to be selected in an ad hoc manner. Why is k=20 chosen? How does varying the number of clusters impact the experimental results?

# Typos etc.

Page 2: Footnote 1 should probably be in main text.

Page 4: "given n training examples without group annotations" --> without *attribute* annotations (since group is attribute + label)

**Summary Of The Paper:**

This paper proposes a rescaling-based postprocessing method that is applied to the output of any machine learning model with the goal of making the predictions more robust. They conduct experiments comparing the proposed method used to postprocess the outputs of several existing robust learning techniques, along with an ERM baseline, on two standard robustness datasets (CelebA, Waterbirds).

**Summary Of The Review:**

I have several concerns with the paper. In addition to the proposed method being neither well-motivated nor well-described, the experimental results also show only very small improvements (in most cases, within the realm of statistical variation) and over baselines that appear not to be tuned at all and perform far below the original published results of the baseline works. I think it is not ready for publication in its current form.

---

> ### Author Response · Authors · 2022-11-10
> **Response to Reviewer TDJz (1/2)**
>
> Thanks for the thorough and constructive review. Following are our responses to your concerns.
>
> ### Motivation of robust scaling
> > why rescaling would be a useful or appropriate technique to advance the community toward this goal is not obvious
>
> Existing group robust optimization approaches have reported worst-group and average accuracies, but do not actively explore the trade-off between them, although they clearly sacrifice the average accuracy to achieve high worst-group accuracy. Our scaling strategy is designed for effective control of the trade-off to understand its exact behavior.
>
> Let assume there is a performance discrepancy between minority and majority classes, where majority classes have higher performance. Then, if we upweight the final prediction score of minority classes, the samples will have more chances to be classified into those classes, thus the accuracy of minority classes increases at the expense of those of majority classes, resulting in a desirable trade-off for group robustness. We demonstrate that our class-specific scaling can identify the optimal point that maximizes any target metrics (ex. worst-group acc) on the trade-off on top of existing models in practice with extensive experiments.
>
> > The equation (3) does not fully define the problem being solved. For example, what is really meany be a "scaling coefficient vector"- does it need to have unit norm? Why do we take arg max in (3)? What are the practical implications of using this method?
>
> For standard classification, we conduct $\arg \max_c(\hat{y})_c$ to predict the target class with the highest classification scores. With our scaling strategy, the scaling coefficient vector $s$ is element-wise multiplied to the final score vector, where $s$ is $C$-dimensional non-negative vector, e.g., $s = (1, ..., 1) \in \mathbb{R}^C$. In other words, we finally find the class with the highest scores after scaling by argmax, i.e., $\arg \max_c(s \odot \hat{y})_c$. Here, argmax indicates classification based on the final scores. We search the scaling coefficient within a sufficient range as described in Appendix D, which takes only a few seconds with CelebA and ResNet-50. We don’t need to do unit norm on the coefficient vectors as we only need to identify which class’s score is the highest after scaling.
>
> ### Results on Waterbirds
> >report much lower numbers than those works on the same dataset
>
> The major difference is that prior works employed **ResNet-50** as the baseline while we adopt ResNet-18. Also, on the Waterbirds dataset, we used the original average accuracy while both GroupDRO and JTT use the **adjusted average accuracy** to report average accuracy; they first calculate the accuracy for each group and then report the weighted average, where the weights correspond to the relative portion of each group in training set. Refer to the discussion regarding this issue (https://github.com/anniesch/jtt/issues/2). Note that both GroupDRO and JTT use the original average accuracy on CelebA.
>
> For more accurate comparisons, we conducted experiments with ResNet-50 and adjusted average accuracy on Waterbirds and got strong reproduced results and RS still achieved meaningful gain on top of them. We updated the new results in Table 2. We followed the same pipeline to search hyperparameters; For JTT, we used 1e-5 of learning rate, 1.0 of weight decay, and 100 of $\lambda_\text{up}$, and we updated the error set every 60 epochs. For Group DRO, we used 1e-5 of learning rate, 0.1 of weight decay, and 0.1 of $\eta$. Please refer to this paper (https://arxiv.org/pdf/2110.14503.pdf) that also reproduces JTT and Group DRO, which reports 85.6(0.2) and 87.1(3.4) of worst-group accuracy with the best hyperparameters, respectively, on Waterbirds dataset. If you run the official implementation with the best hyperparameter configs, you will get similar results. (https://github.com/anniesch/jtt, https://github.com/kohpangwei/group_DRO)
>
> Beside these results, we emphasize that our framework does not care about hyperparameter tuning, because it is **model-agnostic** and works on top of existing models. No matter what the trained model is, it can always find the optimal scaling factor that maximizes the target objective, as shown in the figures and tables. If the original results are already near optimum, then the scaling factor will be close to 1 and at least it does not degrade the performance. This will be analyzed in detail in the following comment (2/2).
>
> >  Group DRO reports much higher Waterbirds worst-group and average accuracies in Table 3
>
> Group DRO takes advantage of the group supervision during training. We also extended our robust scaling that utilizes group supervision, dubbed as attribute-specific scaling (ARS), and addressed in Appendix C. As shown in Table A5, ERM+ARS can match the performance of Group DRO on both datasets.
> Note that, on Waterbirds dataset, ERM+IRS already achieves competitive performance without group annotations to Group DRO.

---

> > ### Author Response · Authors · 2022-11-10
> > **Response to Reviewer TDJz (2/2)**
> >
> > ### Effectiveness of our robust scaling
> > > only a few comparisons show an improvement greater than two standard deviations
> >
> > To analyze the true effectiveness of our robust scaling thoroughly, we measure the performance gain of robust scaling for each run separately, and calculate its mean and variance. This is for eliminating the impact of variance in initial performance. We report the gain results in the **gain row** of Table 1 and 2, colored by blue, which are summarized as:
> >
> > |Gain in WA|ERM|CR|SUBY|LfF|JTT|GDRO|GR|
> > |---|---|---|---|---|---|---|---|
> > |CelebA|+47.7(7.8)|+12.2(7.5)|+15.1(3.0)|+23.2(2.5)|+7.8(3.0)|+1.4(1.0)|+1.4(1.1)|
> > |Waterbirds|+5.3(1.3)|+3.4(1.8)|+3.1(1.4)|+2.6(2.1)|+1.4(0.7)|+1.1(0.8)|+3.2(2.0)|
> >
> > This validates that the robust scaling clearly makes **meaningful performance improvement** in robust accuracy considering standard deviation for all algorithms. Large variance of results mainly comes from different random seeds, but the robust scaling always improves the results for each run.
> >
> > ### Trade-off between robust and average accuracies in Figure 1
> > > There is very weak evidence of a tradeoff here, but the details are far too murky to draw much of a conclusion; if there were not lines drawn on the graphs I would not conclude there was any relationship.
> >
> > In Figure 1, we conducted multiple experimental runs on the CelebA dataset with the same hyperparameter but different random seeds for a fixed 50 epoch, and observed that the worst-group accuracy is negatively correlated to average accuracy. For more thorough analysis, we calculate The Pearson correlation coefficients between worst-group and average accuracies, and achieve **(-0.72, -0.71, -0.80)** for the three methods, all of which clearly have **strong negative correlations**. We updated the coefficient results in the legend of Figure 1.
> >
> > ### Details about Figure 2
> > > what models are used for each method, how they were selected, or give any notion of variability
> >
> > In Figure 2, the dataset is CelebA, and we used the reported results from their original papers. (So it reflects the best hyperparameter configuration for each method.). Figure 2 presents that even the ERM baseline can achieve competitive robust accuracy to most of existing debiasing approaches using our RS and IRS, though the initial robust accuracy of ERM is very low, which supports our argument that more comprehensive evaluation is needed to evaluate the algorithm thoroughly.
> >
> > ### Feasibility of our scaling strategy
> > > how are we supposed to conclude that any other scaling (s \neq 1) is an improvement based on this plot?
> >
> > In Figure 3, the black marker on the curve indicates the original ERM baseline (s=1). To validate that this robust scaling strategy can work well in practice, we have also visualized the relationship between scaling factors and robust accuracies in Figure 4, which presents that the optimal scaling factors can improve robust accuracy, compared to the original ERM. As well, the curves constructed based on validation and test splits are sufficiently well-aligned to each other, which presents that the optimal scaling factor identified in the validation set can be used in the test set as well to get the final robust prediction. Thus, we can identify any Pareto optimal points on the curves of test split, which maximizes worst-group accuracy, average accuracy, or a linear interpolation of them, using our robust scaling strategy following the description in Section 3.2.
> >
> > ### Ablative study on the number of clusters in IRS
> > >  How does varying the number of clusters impact the experimental results?
> >
> > We conducted the ablative study of the number of clusters K in Figure 6, where the sufficient number of K > 10 gives stable results. We can select the optimal number of K based on the robust coverage results of the validation split, as written in Section 4.3.
> >
> > ### Discussion about Robust Coverage
> > > I think the metric is lacking a thorough discussion or motivation.
> >
> > Robust coverage, which claimed the need throughout the paper, is designed to measure the performance of algorithms in a more comprehensive way considering both robust and average accuracies. Thanks to our robust scaling, we can first identify the full trade-off curve without computational overhead, as visualized in Figure 3. Among the points on the curve, our interests are only the Pareto optimums, located within the green area (Pareto optimal) of Figure 3. Other non-optimal points give both inferior worst and average accuracies than optimal ones, so there is no reason to use non-optimal ones.
> >
> > Thus, it is reasonable to measure the performance based on the trade-off curve from Pareto optimal perspective, which is easily measured by Eq. (4). This measurement is simple and straightforward which finds the Pareto optimum for each threshold of average accuracy and summarizes it to a single scalar value. (On a side note, we would like to note that other reviewer stated the robust coverage is our strong contribution.)

---

> ### Author Response · Authors · 2022-12-01
> **Look forward to hearing your feedback**
>
> Thank you for reviewing our paper, and we revised the manuscripts and provided detailed responses to address your concerns. As the discussion period is just around the corner (only a few days left), we hope you respond to our rebuttal if you have any remaining concerns or questions, or if all your concerns have been resolved well. We would be happy to answer additional follow-up questions.

---

> > ### Author Response · Authors · 2022-12-12
> > **Response to Reviewer TDJz: are there any additional questions or concerns?**
> >
> > We hope that our responses have addressed all the concerns and questions raised in the review. Could you let us know if there are any other questions or concerns?

---

### Official Review · Reviewer_2hCQ · 2022-10-22

**Confidence:** 4
**Correctness:** 3
**Technical Novelty And Significance:** 3
**Empirical Novelty And Significance:** 3
**Recommendation:** 5

**Clarity, Quality, Novelty And Reproducibility:**

The clarity is good. The overall quality is satisfactory but the experiment is not rigorous thus not convincing. The novelty is just at the borderline, since the test-time rescaling is not new [1].

[1] Polina Kirichenko, Pavel Izmailov, and Andrew Gordon Wilson. Last layer re-training is sufficient for robustness to spurious correlations. arXiv preprint arXiv:2204.02937, 2022.



**Strength And Weaknesses:**

Strength:
The paper is clearly written and the proposed method is straightforward. The experiment looks extensive.


Weaknesses:

The reported baselines in this paper are not consistent with their official results. For example, GroupDRO has 93.5% average accuracy and 91.4% worse-case accuracy with ResNet50, but this paper's Table 2 reports a much lower accuracy for GroupDRO with ResNet18. The result of JTT has the same concern. So I cannot say the experiment is convincing to me.  Please show the experiment using the same backbone with baselines.

Only two binary classification datasets are used in the main experiment. It will be more convincing if the method works in more challenging tasks like CivilComments, WildCam or FMoW in WILDS benchmark, which contain more classes and groups. The FairFace with 9 classes are used to show the scalability of robust rescaling, but not compared with any baselines. I would like to see a fair comparison with existing baselines on more challenging and real-world datasets.

The title has a very strong statement that class-specific scaling is all you need, which is not supported by the experiment as least for me.

**Summary Of The Paper:**

The paper proposes a test-time rescaling strategy to address the trade-off between worse-case and average accuracy. The rescaling coefficient is optimized on a validation set to maximize a certain objective function, e.g., the worse-group accuracy. With different rescaling coefficients, the method is claimed to find a Pareto frontier of the two objectives (worse-case and average accuracy). The proposed method is compared with different baselines in the experiment and is shown to be effective on two datasets.

**Summary Of The Review:**

Given the weaknesses, I would like to give a conservative score and encourage the author to do a more rigourous experiment in the next version to show the real effect of robust scaling. Otherwise, I cannot agree with the argument that class-specific scaling is ALL you need.

---

> ### Author Response · Authors · 2022-11-10
> **Response to Reviewer 2hCQ**
>
> Thanks for your constructive and helpful comments about our paper. Here are our answers to the raised issues.
> ### Waterbirds results
> > The reported baselines in this paper are not consistent with their official results.
>
> As you pointed out, the major difference is that prior works employed **ResNet-50** as the baseline while we adopt ResNet-18. Also, on the Waterbirds dataset, we used the original average accuracy while both GroupDRO and JTT use the **adjusted average accuracy** to report average accuracy; they first calculate the accuracy for each group and then report the weighted average, where the weights correspond to the relative portion of each group in training set. Refer to the discussion regarding this issue: https://github.com/anniesch/jtt/issues/2. Please also refer to this paper (https://arxiv.org/pdf/2110.14503.pdf) that reproduces JTT and Group DRO, which reports 85.6(0.2) and 87.1(3.4) of worst-group accuracy, respectively, with the best hyperparameter configurations on Waterbirds dataset.
>
> For more accurate comparison, we updated the results using ResNet-50 with adjusted average accuracy on Waterbirds dataset in Table 2, and we achieved similar baseline results with the reported numbers. To measure the true effectiveness of our robust scaling accurately, we also add the *gain row* in Table 1 and 2, which represents the mean and standard deviation of the performance gain for each run. This is for eliminating the impact of variance in initial performance. Table 1 and 2 validate that our robust scaling can achieve meaningful performance improvement (considering variance) for all datasets and algorithms, including JTT and Group DRO with ResNet-50. Please note also that our framework is model-agnostic and works on top of existing models, so it can find the optimal scaling factor that maximizes the target objective (at least without performance degradation) regardless of their hyperparameters.
>
> ### Novelty
> > The novelty is just at the borderline, since the test-time rescaling is not new [1].
>
> As we also mentioned in the related work section, [1] validates that only re-training the last layer with a balanced validation set is enough to achieve high robust accuracy. However, we don’t think that [1] is a complete test-time rescaling method, because it definitely requires extra training. We emphasize that our framework only needs to evaluate the validation split with different scaling factors to tune the scaling factors, which takes only a few seconds with CelebA and ResNet-50. On the other hand, the larger the size of model architecture and dataset, the greater the computational overhead will be for extra training, compared to ours. In addition, because our scaling strategy is very efficient and effective, it enables us to identify the full landscape of the trade-off curve without computational overhead and measure the comprehensive robust coverage, while [1] seems to be not easily applicable for the comprehensive measurement with full landscape.
>
> [1] Polina Kirichenko, Pavel Izmailov, and Andrew Gordon Wilson. Last layer re-training is sufficient for robustness to spurious correlations. arXiv preprint arXiv:2204.02937, 2022.
>
> ### Additional dataset
> >  It will be more convincing if the method works in more challenging task.
>
> To validate the generalization ability and robustness of our proposed algorithm, we will analyze our frameworks in other datasets and update the results when it is finished.
>
> ### Title
> > The title has a very strong statement that class-specific scaling is all you need, which is not supported by the experiment as least for me.
>
> Thanks for your suggestion. We will tone down the title, like ‘Revisiting Group Robustness with Adaptive Class-specific Scaling’. However, please note that our ERM+IRS achieves competitive performance without group supervision to the group-supervised method, Group DRO, on the Waterbirds dataset in Table 3.
> In addition, we have also proposed another extension of our robust scaling in Appendix C, called attribute-specific robust scaling (ARS), which utilizes group supervision of training dataset, like as Group DRO, for robust scaling. As presented in Table A5, ERM+ARS achieves competitive performance to Group DRO on both datasets.

---

> > ### Author Response · Authors · 2022-11-21
> > **Experimental Results with Additional Datasets**
> >
> > Thanks for your patience with our response. We analyzed our frameworks on top of several baselines in the text classification dataset, CivilComments-WILDS, which consists of 16 groups, and achieved strong results consistently. Please refer to the [Robust Scaling with Additional Datasets and Algorithms](https://openreview.net/forum?id=pkgVPeL9gpX&noteId=MUEc9Pms5Q) for the experimental results and detailed discussions.

---

> > ### Author Response · Authors · 2022-12-01
> > **Look forward to hearing your feedback**
> >
> > Thank you for reviewing our paper, and we revised the manuscripts and provided detailed responses to address your concerns. As the discussion period is just around the corner (only a few days left), we hope you respond to our rebuttal if you have any remaining concerns or questions, or if all your concerns have been resolved well. We would be happy to answer additional follow-up questions.

---

### Official Review · Reviewer_yYTs · 2022-10-22

**Confidence:** 4
**Correctness:** 3
**Technical Novelty And Significance:** 3
**Empirical Novelty And Significance:** 3
**Recommendation:** 6

**Clarity, Quality, Novelty And Reproducibility:**

Clarity:

The paper is well-written and the proposed method is simple yet novel in the spurious correlation literature.

**Strength And Weaknesses:**

Strength:
- The proposed method is simple yet very effective and can be easily added to previous methods.
- The paper proposes a new metric to both evaluate average accuracy and worst-group accuracy.

Weakness:
- It is clear to me that the proposed method is able to maximize the robust accuracy, it remains unclear to me why it can achieve better balance between robust accuracy and worst-group accuracy than the other methods.
- The method may heavily rely on the validation set to work and requires access to the group information of the validation set which sometimes is not necessarily available.
- The method might also need a large validation set to work as the validation set is used to estimate the scaling factor and the centroid. It would be nice if an ablation study can be performed for different sizes of the validation set.


Question:
- Wondering instead of searching the scaling factor, can it be learned?
- does adding a bias factor be more effective than scaling factor along?

**Summary Of The Paper:**

This paper suggests rescaling the output probability to achieve group robustness. The idea is to find the scale factor using the unbiased validation set to minimize the balanced group accuracy on the validation set. Once the scale factor is found, it is applied during the test as a post-process module to the test prediction.

**Summary Of The Review:**

The paper proposes a simple yet novel technique to post-process the network prediction to achieve group robustness. The method is well-motivated and experiments are conducted to validate its effectiveness.

---

> ### Author Response · Authors · 2022-11-14
> **Response to Reviewer yYTs [1/2]**
>
> We thank your positive and meaningful comments and suggestions. Following is our answer to your comments.
>
> ### Robust scaling
> > It is clear to me that the proposed method is able to maximize the robust accuracy, it remains unclear to me why it can achieve better balance between robust accuracy and worst-group accuracy than the other methods.
>
> Our framework aims to identify the inherent trade-off between robust and average accuracies of algorithms by adopting simple class-specific scaling. We do not claim that our method gives better balance between two kinds of accuracies, but argue that each existing algorithm has its own trade-off curve and our robust scaling 1) can find the optimal points that maximize the target objective (e.g. worst-group acc) on the trade-off curve and 2) enables to measure its own trade-off curve in a comprehensive way. While performing robust scaling, the trade-off curve remains the same.
>
> If your question means why ERM achieves competitive trade-off results to some other baselines, we believe this is partly because many existing approaches conduct sample reweighting to focus on the minor groups (refer to related work section), which may give similar effects to our class-specific rescaling as a result.
> Note that some other papers [1, 2] also argue that simple group reweighting or subsampling achieve competitive robust accuracy to other state-of-the-art approaches. However, different from these works, our framework is much more efficient in that it does not require any extra training but yet achieves meaningful performance improvement in robust accuracy.
>
> [1] Idrissi et al., "Simple data balancing achieves competitive worst-group-accuracy", CLeaR 2022
>
> [2] Polina et al., "Last layer re-training is sufficient for robustness to spurious correlations", arXiv 2022
>
>
> ### Ablative study on the validation set
> > It would be nice if an ablation study can be performed for different sizes of the validation set.
>
> Our robust scaling strategies need a held-out validation set to identify the optimal scaling factors, like as other existing approaches for early stopping and hyperparameter tuning for high robust accuracy. (Note that early stopping is essential to achieve high robust accuracy in existing approaches.)
> To validate the robustness of our frameworks, we conduct the ablation study by varying the size of validation set on the CelebA dataset in the following table.
> |Method|Valid set size|Worst-group|(gain)|Unbiased|(gain)|
> |------|---|---|---|---|---|
> |ERM|100%| 34.5(6.1)|- | 77.7(1.8) |-|
> |+RS|100%| 82.8(3.3) |**+48.3**| 91.2(0.5) |**+13.5**|
> |+RS|50%| 83.3(3.7) |**+48.8**| 91.5(0.9) |**+13.8**|
> |+RS|10%| 82.4(4.3) |**+48.0**| 91.4(0.8) |**+13.7**|
> |+RS|1%| 79.2(10.3) |**+44.7**| 90.8(2.2) |**+13.1**|
> |+IRS|100%|88.7(0.9) |**+54.2**| 92.0(0.3) |**+14.3**|
> |+IRS|50%|86.9(2.0) |**+52.4**| 91.8(0.4) |**+14.1**|
> |+IRS|10%|84.4(6.3) |**+50.0**| 91.4(1.0) |**+13.7**|
> |+IRS|1%|60.4(14.4) |**+25.9**| 85.8(3.2) |**+8.0**|
>
>
>
>
> We vary the size of validation set by {100%, 50%, 10%, 1%}.
> As shown in the table, with only 10% or 50% of the validation set, both our robust scaling (RS) and instance-wise robust scaling (IRS) achieve almost competitive performance with the results using the full size of validation set.
> Surprisingly, even only 1% of the validation set is enough for RS to achieve sufficiently high robust accuracy, but inevitably entails a large variance of results.
> On the other hand, IRS suffers from limited performance gain when only 1% of the validation set is available.
> This is mainly because IRS takes advantage of feature clustering on the validation set, which needs some validation samples.
> Nevertheless, our robust scaling strategies can achieve meaningful performance improvement with a limited number of validation samples for all cases, which validates the robustness of our method.
> Thanks for suggesting this important ablative study. We added this in our appendix, and will move to the main paper.

---

> > ### Author Response · Authors · 2022-11-14
> > **Response to Reviewer yYTs [2/2]**
> >
> >
> > ### Learnable scaling factor
> > > Wondering instead of searching the scaling factor, can it be learned?
> >
> > As you suggested, scaling factors can also be learned using gradient descent. However, because our scaling factor is applied on top of the final classification layer, it is almost equivalent to a single layer training, which is unlikely to benefit from learning. We empirically found that a simple grid search or other search algorithms (e.g. L-BFGS) are more efficient and effective than gradient descent; our robust scaling (with a simple search) only takes a few seconds on the CelebA dataset with ResNet-50. Note that temperature scaling [3] also employs the grid search or other search algorithms rather than gradient descent to find the optimal temperature for the same reason.
> >
> > [3] Guo et al., "On calibration of modern neural networks.", ICML 2017
> >
> > ### Bias vs Scaling factor
> > > does adding a bias factor be more effective than scaling factor along?
> >
> > Because our optimal factor is tuned using a search algorithm instead of being learned, there is no big difference between using bias factors, scaling factors, or both, and any of them can be used to search the optimal factors.

---

> ### Author Response · Authors · 2022-12-01
> **Look forward to hearing your feedback**
>
> Thank you for reviewing our paper, and we revised the manuscripts and provided detailed responses to address your concerns. As the discussion period is just around the corner (only a few days left), we hope you respond to our rebuttal if you have any remaining concerns or questions, or if all your concerns have been resolved well. We would be happy to answer additional follow-up questions.

---

> > ### Comment · Reviewer_yYTs · 2022-12-04
> > **Still leaning towards acceptance**
> >
> > Dear AC, and other reviewers,
> >
> > I have reread the updated version of the paper and the authors' response. I still believe the paper provides many interesting insights, as also mentioned by Reviewer 3mfo -- e.g. the trade-off between worst and avg accuracy. The simplicity and effectiveness of the proposed method also give me a positive impression. I also believe the paper is well-written and of good quality.
> >
> > I also agree with other reviewers that the title "CLASS-SPECIFIC SCALING IS ALL YOU NEED" is a little overclaimed, and the link between the illustrated "trade-off" and the proposed method is a little weak. Regarding the experiments, I believe the author did a good job explaining the lower performance of the baselines. Moreover, following the  ICLR reviewer guide, the answer to the question "If a submission does not achieve state-of-the-art results, is that grounds for rejection?", I think the experiments, even though they don't achieve SoTA on some datasets, are sufficient to support the effectiveness of the method.
> >
> > Therefore, for overall consideration, I would still keep my original score unchanged and recommend accepting the paper.

---

> > > ### Author Response · Authors · 2022-12-04
> > > **Response to Reviewer yYTs**
> > >
> > > We appreciate your time to read all our revision, responses, and extensive discussions with other reviewers. As we mentioned in another response, we will tone down the title of our paper, and also reflect all other suggestions and our responses in the final copy. Thanks for your careful feedback and feel free to let us know if you have any other questions.

---

### Official Review · Reviewer_3mfo · 2022-10-27

**Confidence:** 4
**Correctness:** 3
**Technical Novelty And Significance:** 2
**Empirical Novelty And Significance:** 4
**Recommendation:** 8

**Clarity, Quality, Novelty And Reproducibility:**

The writing is easy to read, but plots could be significantly better and the results better presented:
- I did not understand what the takeaway from figure 1 was. I would suggest having figure 2 as the main figure 1.
- Which dataset is presented in Figure 1?
- Figure 4 can safely be moved to the appendix - added value is not that much.
- When reading, it was often unclear whether the results with RS referred to IRS or to RS. This is since IRS is presented before any experiments, so in the first part of the experiments it was unclear whether the results were for RS or IRS. To remedy this, one suggestion would be to introduce the IRS method *after* the first set of RS experiments.
- Why are table 4 experiments performed on a completely random dataset used nowhere else in the paper?
- Figure 5 bottom can be safely removed - it does not add any additional info to Figure 5 top. I would suggest using the space to draw the scaling curves for GDRO, JTT, and other important baselines instead.
- There is no discussion of Table 2 in section 4.2.

**Strength And Weaknesses:**

Strengths:
- the proposed method (RS) and its extension (IRS) are both simple and effective
- the robust coverage method is a nice way to summarize the worst-avg accuracy tradeoff and is a strong contribution
- the related works section is written very well.

Weaknesses:
- Evaluation on more datasets would help to strengthen the analysis. In particular, the BalancingGroups codebase (https://github.com/facebookresearch/BalancingGroups) would be a great point for this to add 2 text classification tasks.
- Following the above work, this work should also compare against subsampling baselines (SUBG/SUBY).
- The results are often not presented in the clearest way (elaborated in more detail below).
- In general, it is unclear to me whether this paper thinks RS or IRS should be the main method (there is also a discussion of ARS in the appendix, which seems to have very strong results but confusingly was not even discussed in the main text). It seems that IRS does have strong results on Waterbirds and should be highlighted more - in fact, I would suggest adding IRS to figure 2 and presenting all results (tables 1 & 2) with both RS and IRS. (As a side note, it is also unclear why IRS was not evaluated on GDRO/JTT etc in table 3). This point may seem small, but it has a significant impact on how the results are interpreted overall.

**Summary Of The Paper:**

This work presents a simple post-training strategy to balance the worst-group and average-case accuracies of a classifier. In addition, it proposes a new metric to summarize this tradeoff. An extension on the simple method is also proposed to achieve better results. The proposed method outperforms several existing baselines and helps to put prior work on group robustness in context.

**Summary Of The Review:**

This paper presents some important insights regarding the tradeoff between worst and average accuracy. It presents a simple method to allow trading off the two. At the same time, this work lacks some experimental rigor (only 2 datasets and missing subsampling baseline) and clarity in exposition (the RS & IRS results are difficult to compare to each other). If these last two points were done well, there would be a nice takeaway along two directions: 1) worst and avg accuracy can be traded off for all models, and 2) RS & IRS are two simple methods that do this with strong empirical performance. Due to the limitations outlined, however, I can only have confidence and clarity in takeaway #1. This is the reason for my rating.

------------------------ UPDATE AFTER REBUTTAL --------------------

My main criticisms with this work are around clarity and the small number of datasets. The authors have addressed clarity somewhat, and importantly have added results on two more datasets - CivilComments and FMOW. The results on CivilComments are just as impressive/notable as those on CelebA. The results on FMOW are a bit muddied, which is due to ERM performing worse than GDRO, so the story here is a bit nuanced but I am glad the authors included the results nonetheless. Given that evaluation has been much improved, I have updated my rating of this paper to accept. That being said, I would like to see a couple things in the camera ready:
- Full evaluation results of all the baselines on the new datasets.
- A figure or table within the main text or appendix that averages the numbers across all datasets (to provide a wholistic view of the method).

---

> ### Author Response · Authors · 2022-11-17
> **Response to Reviewer 3mfo [1/3]**
>
> We appreciate your detailed and helpful review and comments throughout the paper. We will reflect your suggestions at our best, and here’s our answers to your questions.
>
>
> ### Additional datasets/baselines
> > Evaluation on more datasets would help to strengthen the analysis. Following the above work, this work should also compare against subsampling baselines (SUBG/SUBY).
>
> Thanks for your suggestion. Our framework turns out to be very effective in two standard benchmarks, and we agree its effectiveness will be strengthened with additional experiments using different baselines/datasets.
> To validate the robustness of our framework, we first analyzed our robust scaling strategies on top of two additional baselines from [1], SUBG and SUBY, on CelebA and Waterbirds datasets.
> Table A9 in the appendix presents the full experimental results, some of which are summarized as:
> |Dataset|Method|Worst Cover.|Unbiased Cover.|Worst Acc.|Unbiased Acc.|Avg Acc.|
> |---|---|---|---|---|---|---|
> |Waterbirds|SUBG|-|-|86.5(0.9)|88.2(1.2)|87.3(1.1)|
> ||SUBG+RS|80.6(2.0)|82.3(2.0)|87.1(0.7)|**88.5(1.2)**|91.3(0.4)|
> ||SUBG+IRS|**82.2(0.8)**|**84.1(0.8)**|**87.3(1.3)**|88.2(1.2)|**93.5(0.4)**|
> |Waterbirds|GR|-|-|86.1(1.3)|89.3(0.9)|95.1(1.3)|
> ||GR+RS|83.7(0.3)|86.8(0.7)|**89.3(1.3)**|92.0(0.7)|95.4(1.3)|
> ||GR+IRS|**84.8(1.7)**|**87.4(0.4)**|89.1(0.8)|**92.2(1.0)**|**95.6(0.8)**|
> | | | | | | | |
> |Waterbirds|SUBY|-|-|72.8(4.1)|84.9(0.4)|93.8(1.5)|
> ||SUBY+RS|72.5(1.0)|81.2(1.4)|75.9(4.4)|86.3(0.9)|95.5(0.2)|
> ||SUBY+IRS|**78.8(2.7)**|**85.9(1.0)**|**82.1(4.0)**|**89.1(0.9)**|**96.2(0.6)**|
> |Waterbirds|CR|-|-|76.1(0.7)|89.1(0.7)|97.1(0.5)|
> ||CR+RS|73.6(2.3)|82.0(1.5)|79.4(2.4)|89.4(1.0)|**97.5(0.3)**|
> ||CR+IRS|**84.2(2.5)**|**88.3(1.0)**|**88.2(2.7)**|**92.1(0.7)**|97.4(0.2)|
> | | | | | | | |
> |CelebA|SUBG|-|-| 87.8(1.2) | 90.4(1.2) | 91.9(0.3) |
> ||SUBG+RS| 83.6(1.6) | 87.5(0.7) | 88.3(0.7) | 90.9(0.5) | 93.9(0.2)|
> ||SUBG+IRS| **84.5(0.8)** | **87.9(0.1)** |**88.7(0.6)** | **91.0(0.3)** | **94.0(0.2)**|
> |CelebA|GR|-|-| 88.6(1.9) | 92.0(0.4) | 92.9(0.8) |
> ||GR+RS| 86.9(0.4) | 88.4(0.2) | **90.0(1.6)** | 92.4(0.5) | 93.8(0.4)|
> ||GR+IRS| **87.0(0.2)** | **88.6(0.2)** |**90.0(2.3)** | **92.6(0.6)** | **94.2(0.3)**|
>
> To implement SUBG and SUBY, we tune the hyperparameters by conducting the grid search on learning rate in {$10^{-3}, 10^{-4}, 10^{-5}$} and weight decay in {$1.0, 0.1, 10^{-2}, 10^{-4}$}.
> As shown in the table, our framework still manages to achieve meaningful performance improvement with two additional baselines; the original robust scaling (RS) can identify the optimal points that maximize each target objective and instance-wise robust scaling (IRS) improve the overall trade-off curve and further improves the performance of target objective.
>
> To be more specific, we observe that reweighting baseline (CR, GR) and subsampling baselines (SUBY, SUBG) achieves similar results on the CelebA dataset.
> However, there exists some noticeable performance gaps on the Waterbirds dataset.
> For example, SUBG (group subsampling) achieves competitive robust accuracy with GR (group reweighting), but at the same time, SUBG has much lower average accuracy than that of GR, which is also related to its low robust coverage and low robust accuracy (after scaling) results.
> This is mainly because SUBG drops a large portion of training samples to make all groups have the same size with the smallest group. Note that the number of training samples for each group in Waterbirds are {3498, 56, 184, 1057} and the smallest group only has 56 samples, so SUBG drops 95.3% of training samples, resulting in significant performance degradation in average accuracy.
> Although this subsampling generally helps to achieve high robust accuracy (without scaling), it degrades the overall trade-off and consequently hinders the benefits of robust scaling.
> Note that GR outperforms SUBG in terms of all worst-group, unbiased, and average accuracies after scaling.
> Similar tendency is also found in SUBY, but not as severe as SUBG, because each class has {1113, 3682} samples and SUBY drops 53.6% of training samples.
>
> On the other hand, the subsampling baselines achieve stable results on the CelebA dataset.
> Because the smallest group has a sufficient number of samples (1387), SUBG does not suffer from a lack of training data, resulting in no significant performance gap between reweighting and subsampling baselines.
>
> (Continued on the next comment)

---

> > ### Author Response · Authors · 2022-11-17
> > **Response to Reviewer 3mfo [2/3]**
> >
> > (Continued from the previous comment)
> >
> > Based on the comprehensive evaluation results, it can be claimed that, if the dataset is small-scale and highly imbalanced, then reweighting based methods are more effective than subsampling baselines.
> > We believe these results consistently demonstrate the effectiveness of our framework and support our main claim; considering only the robust accuracy is incomplete and comprehensive evaluation is needed to understand the exact behavior.
> > Thanks for introducing important baselines. We added [1] in the related work section, and will move this results to the main paper and add more detailed discussion. Also, we are going to conduct experiments on additional datasets/tasks, and report the results as soon as it is finished.
> >
> > [1] Idrissi et al., "Simple data balancing achieves competitive worst-group-accuracy", CLeaR 2022
> >
> >
> >
> > ### Flow
> > Thanks for detailed and careful comments about the paper structure. We will reflect your suggestions at our best. Please understand that we will update the overall structure after the initial discussion is over (to avoid the confusion in numbering).
> >
> >
> > >  it is unclear to me whether this paper thinks RS or IRS should be the main method. It seems that IRS does have strong results on Waterbirds and should be highlighted more.
> >
> > We agree with your suggestion that IRS should be highlighted more as it provides strong and consistent results that can overcome the trade-off of each algorithm.
> > However, some readers may only focus on the performance improvement by IRS, rather than our main claim regarding the inherent trade-off for each algorithm, which can be easily controlled with a simple scaling.
> > This is why we separate Table 1,2  and Table 3. Table 1 and 2 evaluate the existing algorithms in a comprehensive way, which provides a novel perspective that sometimes give different tendency than the original robust accuracy results.
> > Table 3 (and Table A9) presents that IRS can overcome the overall trade-off and improve the performance of both target objectives furthermore.
> > That being said, we agree with your concern and will revise the overall structure thoroughly for a clear understanding. Thanks for pointing this out.
> >
> >
> > >  in the first part of the experiments it was unclear whether the results were for RS or IRS. To remedy this, one suggestion would be to introduce the IRS method after the first set of RS experiments.
> >
> > RS denotes the original robust scaling throughout the paper, and we explicitly state IRS when referring to IRS. Sorry for the confusion, and we will clarify and modify the overall flow throughout the paper considering all your suggestions.
> >
> >
> > > I would suggest adding IRS to figure 2 and presenting all results (tables 1 & 2) with both RS and IRS.
> >
> > We visualize the ERM+IRS results in Figure 2 and conduct the experimental results with both our RS and IRS on top of existing baselines in Table A9.
> >
> >
> > ### ARS
> > > there is also a discussion of ARS in the appendix, which seems to have very strong results but confusingly was not even discussed in the main text
> >
> > Thanks for noticing the strong results of ARS in the appendix. We initially included ARS in our main paper, but moved it to the appendix because ARS can give the reader the impression that additional supervision is essential to achieve high performance. (Note that our RS and IRS are already achieving meaningful performance gains without group supervision.) We will elaborate the experimental results of ARS.

---

> > > ### Author Response · Authors · 2022-11-17
> > > **Response to Reviewer 3mfo [3/3]**
> > >
> > > ### Detailed comments
> > >
> > > >  I did not understand what the takeaway from figure 1 was. I would suggest having figure 2 as the main figure 1.
> > >
> > > Figure 1 presents the motivation of our scaling strategy that there is a clear trade-off between robust and average accuracies from the same algorithm with different random seeds. We will set Figure 2 as our first figure. Both figures are based on the CelebA dataset.
> > >
> > >
> > >
> > > > Figure 4 can safely be moved to the appendix - added value is not that much.
> > >
> > >
> > > Figure 4 demonstrates the feasibility of our framework in practice. To be specific, it visualizes the relationship between scaling factors and robust accuracies in validation and test splits. Because the scaling results based on validation and test splits are sufficiently close to each other, we can use the optimal scaling factors identified in the validation split to get the final robust prediction in the test split. (On a side note, one reviewer asked the question regarding this feasibility issue and we answered with Figure 4.) As well, it also presents the existence of global optimums along the curve, which can be easily found. Note that Figure 3 itself does not fully address the above feasibility concerns.
> > >
> > >
> > >
> > >
> > > > Why are table 4 experiments performed on a completely random dataset used nowhere else in the paper?
> > >
> > > Table 4 is employed to validate the scalability of our framework, which has a total of 63 groups (7 class, 9 bias groups), but some other existing approaches do not give stable results on this dataset. We will add additional experiments with various baselines to validate the generalization ability of our framework.
> > >
> > > > Figure 5 bottom can be safely removed - it does not add any additional info to Figure 5 top.
> > >
> > > Figure 5 visualizes the Pareto frontiers to help readers who are not familiar with. We agree with your opinion and will move the bottom subfigures to the supplementary file.
> > >
> > >
> > > > There is no discussion of Table 2 in section 4.2
> > >
> > > We added more explanations about Table 2 in Section 4.2. Thanks for pointing that out.
> > >
> > > ### Summary
> > > >  This paper presents some important insights regarding the tradeoff between worst and average accuracy. It presents a simple method to allow trading off the two. At the same time, this work lacks some experimental rigor (only 2 datasets and missing subsampling baseline) and clarity in exposition (the RS & IRS results are difficult to compare to each other).
> > >
> > > For experimental rigor, we analyzed additional subsampling baselines (SUBG, SUBY) with our frameworks, and we will add more results with additional datasets. For clarity, we reported both RS & IRS results on existing methods in Table A9 for a comprehensive view. We will thoroughly revise the paper for better understanding.

---

> > > > ### Comment · Reviewer_3mfo · 2022-11-17
> > > > **clarity much improved, but additional datasets are key**
> > > >
> > > > Thanks to the authors for running the additional baselines; they greatly improve the strength of the results. Also, thanks to the authors for improving the clarity regarding the figures and tables.
> > > >
> > > > My main criticism of this paper is that results are only shown on 2 datasets - celeba and waterbirds - neither of which are great datasets. Adding some text classification tasks (such as those in wilds) or some other vision datasets (such as DomainNet, or GeoYFCC for example) would greatly improve my confidence in the generality of the method and results. I understand this is not doable within the rebuttal time-frame, so perhaps it is better suited for resubmission.
> > > >
> > > > Another drawback is that the clarity in presentation can still be improved (though it has already gotten better). The strength of this work is that RS (and even IRS) are dead-simple post-training tools to trade off robust and average accuracy. It's nice you can apply them to any existing method as well, but the main comparisons should be between ERM + RS and all the other robust learning methods - here the takeaway would be something like, it's better than all other methods that don't use group labels and slightly worse than the methods that do. But the results presentation is currently not structured in this way.
> > > >
> > > > In spite of these drawbacks, the main message of this paper that 1) robust and average accuracy lie on a tradeoff curve, and that 2) RS is one simple way to navigate this tradeoff, is great. The authors also introduce a coverage metric, which would be great to have wide adoption in usage for this type of work. Thus I am split on my opinion of this paper. I will keep my score as is. If there were a couple more datasets added, I would vote to accept without reservation.

---

> > > > > ### Author Response · Authors · 2022-11-20
> > > > > **Response to Reviewer 3mfo**
> > > > >
> > > > > We appreciate your thoughtful and encouraging comments. We are now running our frameworks on other datasets to validate the generalization ability and will report the results soon. We will also revise Table 3 (and Table A9) to facilitate comparisons between ERM+RS/IRS and other baseline models.

---

> > > > > ### Author Response · Authors · 2022-11-21
> > > > > **Experimental Results with Additional Datasets**
> > > > >
> > > > > Thanks for your patience with our response. We analyzed our frameworks on top of several baselines in the text classification dataset, CivilComments-WILDS, which consists of 16 groups, and achieved strong results consistently. Please refer to the [Robust Scaling with Additional Datasets and Algorithms](https://openreview.net/forum?id=pkgVPeL9gpX&noteId=MUEc9Pms5Q) for the experimental results and detailed discussions.

---

> > > > > > ### Author Response · Authors · 2022-12-07
> > > > > > **Results with FMoW Dataset**
> > > > > >
> > > > > > We report additional experiments on another multi-class dataset, **FMoW**, which consists of 62 classes and 80 attribute groups (16 years $\times$ 5 geographical regions). We evaluated out-of-distribution performance for robust and average accuracies, where training, validation and test sets are collected from different years and group is defined by geographical regions. Following previous work [1] , we adopted DenseNet-121 as the backbone architecture and tuned the learning rate in {$10^{-4}, 10^{-5}$} and weight decay in {$0, 10^{-4}$}. We report the experimental results on the FMoW dataset as follows.
> > > > > >
> > > > > > |Method|Worst Acc.|(gain)|Unbiased Acc.|(gain)|Average Acc.|(gain)|Worst Cover.|Unbias Cover.|
> > > > > > |---|---|---|---|---|---|---|---|---|
> > > > > > |ERM|34.5(1.4)|-|51.7(0.5)|-|52.6(0.8)|-|-|-|
> > > > > > |ERM + RS|35.7(1.6)|**+1.2(0.4)**|52.3(0.3)|**+0.6(0.3)**|53.1(0.8)|**+0.6(0.3)**|32.9(0.4)|39.4(1.3)|
> > > > > > |ERM + IRS|36.2(1.4)|**+1.7(0.3)**|52.4(0.2)|**+0.7(0.4)**|53.4(0.9)|**+0.8(0.4)**|35.1(0.2)|40.2(1.1)|
> > > > > > |GroupDRO|33.7(2.0)|-|50.4(0.7)|-|52.0(0.4)|-|-|-|
> > > > > > |GroupDRO + RS|36.0(2.4)|**+2.3(0.4)**|50.9(0.6)|**+0.4(0.4)**|52.4(0.2)|**+0.5(0.2)**|30.8(1.8)|38.2(0.7)|
> > > > > > |GroupDRO + IRS|36.4(2.3)|**+2.7(0.4)**|51.1(0.3)|**+0.7(0.5)**|52.7(0.2)|**+0.7(0.2)**|34.1(0.8)|40.7(0.5)|
> > > > > > |GR|31.4(1.1)|-|49.0(0.9)|-|50.1(1.3)|-|-|-|
> > > > > > |GR + RS|35.5(0.4)|**+4.2(0.7)**|49.8(0.7)|**+0.8(0.3)**|50.7(1.2)|**+0.6(0.1)**|30.2(1.2)|37.7(0.6)|
> > > > > > |GR + IRS|35.7(0.9)|**+4.4(0.4)**|50.1(0.6)|**+1.1(0.3)**|50.8(1.4)|**+0.7(0.1)**|31.7(1.0)|38.9(2.1)|
> > > > > >
> > > > > >
> > > > > > Unlike CivilComments dataset, the baseline performances of Group DRO and GR are worse than ERM on this FMoW dataset, even with group supervision. Note that, unlike other group robust optimization approaches, our frameworks (RS and IRS) achieve meaningful performance improvement in all kinds of accuracies without any degradation. Because our framework is model-agnostic, it can also be applied to Group DRO and GR, improving their performances with considerable margins. This consistently validates the strengths and robustness of our framework in more challenging datasets. We will elaborate the results with more detailed discussions in the final copy of our paper.
> > > > > >
> > > > > > [1] Koh et al., WILDS: A Benchmark of in-the-Wild Distribution Shifts, arXiv 2020

---

> > > > > > > ### Comment · Reviewer_3mfo · 2022-12-09
> > > > > > > **Additional experiments appreciated, score increased**
> > > > > > >
> > > > > > > I apologize for the delayed response. I thank the authors for running additional experiments in the meantime; they are greatly appreciated.
> > > > > > >
> > > > > > > My main criticisms with this work were around clarity and the small number of datasets. The authors have addressed clarity somewhat, and importantly have added results on two more datasets - CivilComments and FMOW. The results on CivilComments are just as impressive/notable as those on CelebA. The results on FMOW are a bit muddied, which is due to ERM performing worse than GDRO, so the story here is a bit nuanced but I am glad the authors included the results nonetheless. Given that evaluation has been much improved, I have updated my rating of this paper to accept. That being said, I would like to see a couple things in the camera ready:
> > > > > > > - Full evaluation results of all the baselines on the new datasets.
> > > > > > > - A figure or table within the main text or appendix that averages the numbers across all datasets (to provide a wholistic view of the performance of each method).
> > > > > > >
> > > > > > > The clarity of the writing can still be improved overall, but those comments I have already detailed previously. I have raised my score, conditional on the above two changes being incorporated for the camera ready - authors, please acknowledge this.

---

> > > > > > > > ### Author Response · Authors · 2022-12-09
> > > > > > > > **Response to Reviewer 3mfo**
> > > > > > > >
> > > > > > > > We thank your acknowledgement of our responses and discussions for comprehensive analysis with additional datasets/algorithms and clarity. We will reflect all your comments, especially regarding the presentation to facilitate the comparisons between RS/IRS and other robust optimization approaches and also two final suggestions (full baseline results and average across datasets). We will thoroughly re-write the overall structure for better understanding. Please let us know if you have any other suggestions and we will do our best to address them in the final version of our paper.

---

### Author Response · Authors · 2022-11-21
**Robust Scaling with Additional Datasets and Algorithms**

### Additional dataset
We analyzed our frameworks on top of several existing baselines in the text classification task on the **CivilComments-WILDS** dataset [1]. This dataset is large-scale (270K of training images, 45K of validation images, 130K of test images) and has multiple values of bias attribute (male, female, LGBTQ, christian, muslim, other_religions, black, white) with a total of 16 groups, which is more challenging than CelebA and Waterbirds. We report the experimental results as follows.

|Method|Worst Acc.|(gain)|Unbias Acc.|(gain)|Avg Acc.|(gain)|Worst Cover.|Unbias Cover.|
|---|---|---|---|---|---|---|---|---|
|ERM|54.5(6.8)|-|75.0(1.2)|-|92.3(0.4)|-|-|-|
|ERM + RS|65.5(1.2)|**+11.0(2.5)**|78.6(1.5)|**+3.7(2.4)**|92.5(0.3)|**+0.2(0.1)**|57.2(5.1)|70.9(1.5)|
|ERM + IRS|67.0(2.3)|**+12.5(2.7)**|78.8(1.1)|**+3.8(1.7)**|92.5(0.3)|**+0.2(0.1)**|59.2(5.3)|71.2(2.3)|
|GroupDRO|67.7(0.6)|-|78.4(0.6)|-|90.0(0.1)|-|-|-|
|GroupDRO + RS|68.8(0.7)|**+1.1(0.5)**|78.8(0.4)|**+0.4(0.3)**|90.5(0.2)|**+0.5(0.3)**|60.6(0.6)|71.5(0.3) |
|GroupDRO + IRS|69.6(0.4)|**+1.9(0.6)**|78.8(0.5)|**+0.4(0.6)**|90.8(0.3)|**+0.8(0.3)**|62.1(0.7)|71.9(0.2) |
|GR|64.7(1.1)|-|78.4(0.2)|-|87.2(1.0)|-|
|GR+RS|66.0(0.5)|**+1.3(0.6)**|78.5(0.1)|**+0.1(0.1)**|87.9(0.8)|**+0.7(0.3)**|59.0(2.8)|69.8(1.0)|
|GR+IRS|66.2(0.4)|**+1.6(0.7)**|78.6(0.1)|**+0.2(0.2)**|88.4(0.6)|**+1.2(0.6)**|59.7(1.6)|70.1(0.7)|



We adopted DistilBert as the backbone architecture for all baselines following the previous paper [1]. To reproduce each algorithm, we conduct the hyperparameter search on learning rate in {$10^{-5}, 10^{-6}$} and weight decay in {$10^{-2}, 10^{-3}, 10^{-4}$}.
As shown in the above table, our frameworks still work well on top of several baselines in this dataset. This consistently validates that our framework is **model-agnostic**; the original robust scaling (RS) can find the optimal scaling factor that maximizes each target objective and our instance-wise robust scaling (IRS) further improves the performance for all target metrics, regardless of model algorithms.

To be more specific, Group DRO already provides high robust accuracy, but our frameworks still can further improve the performance for all metrics (*worst-group*, *unbiased*, and *average* accuracy). Although group reweighting baseline (GR) achieves much higher initial robust accuracy than ERM, our robust scaling (RS, IRS) enables ERM to catch up the performance of GR. Note that ERM+IRS outperforms both Group DRO and GR in *average* accuracy while achieving competitive *worst-group* and *unbiased* accuracies to them, even without the group supervision of training samples as well as extra training. This validates the excellence and robustness of our frameworks in a more challenging dataset/task. We will elaborate the experimental results on additional datasets with detailed discussions in the final version of our paper.

We also analyzed our frameworks on **FMoW dataset** and please refer to [Results with FMoW dataset](https://openreview.net/forum?id=pkgVPeL9gpX&noteId=0TXocRYg5o) for the results.

[1] Koh et al., WILDS: A Benchmark of in-the-Wild Distribution Shifts, arXiv 2020


### Additional baseline algorithms

We analyzed our robust scaling strategies with two additional baselines from [2], SUBG and SUBY, on the CelebA and Waterbirds datasets, and achieved consistent and robust results. Please refer to [Response to Reviewer 3mfo [1/3]](https://openreview.net/forum?id=pkgVPeL9gpX&noteId=6p8zBwD4vmE) for detailed discussions, and **Table A9** in the appendix of our paper for more comprehensive results on both datasets.

[2] Idrissi et al., "Simple data balancing achieves competitive worst-group-accuracy", CLeaR 2022

---

### Author Response · Authors · 2022-11-21
**General Response and Revision Summary**

Dear Reviewers,

We thank you all for your constructive comments and suggestions, which greatly improve the quality of our paper.  We have addressed all the concerns and questions from the reviewers in the response of each reviewer and the general response (for additional datasets/algorithms). If you have any remaining concerns then feel free to let us know. We would be happy to continue the discussion if there are additional questions.


We first emphasize the contributions and strengths of our paper generally agreed by the reviewers:
- We address the inherent trade-off between robust and average accuracies in existing group robust algorithms, which has not been actively explored yet.
- Our robust scaling strategy is simple and straightforward yet very effective, which can be easily added to any existing algorithms with no extra training; it can identify the optimal points that maximize the target metric on the trade-off curve, regardless of baseline models.
- The robust coverage is a straightforward and nice way to summarize the trade-off considering both robust and average accuracies without computational overhead.
- Our advanced scaling framework, instance-wise robust scaling (IRS), provides strong performance improvement by taking advantage of feature clustering without group supervision.


We summarize the key points from our additional experiments and responses:
- We analyzed our frameworks with additional algorithms (SUBG, SUBY) and different dataset/task (CivilComments-WILDS for text classification and FMoW for multi-class classification), and achieved strong results consistently. This strengthens the generalization ability of our framework on various datasets/algorithms. [Additional Datasets/Algorithms](https://openreview.net/forum?id=pkgVPeL9gpX&noteId=MUEc9Pms5Q)
- We reported the mean and variance of performance gain to analyze the true effectiveness of our frameworks on various baselines in Table 1 and 2 of our paper, which presents meaningful performance improvements for all cases.
- The ablation study on the size of the validation set demonstrates that our frameworks can work well with only some validation samples. [Response to Reviewer yYTs [1/2]](https://openreview.net/forum?id=pkgVPeL9gpX&noteId=kaR_UCWXoKz)
- Our frameworks provide a novel perspective to help understand the exact behavior of algorithms beyond comparing only the robust accuracy. [Response to Reviewer 3mfo [1/3]](https://openreview.net/forum?id=pkgVPeL9gpX&noteId=6p8zBwD4vmE)
  - For example, on the Waterbirds dataset, although subsampling baselines (SUBG, SUBY) achieve competitive robust accuracy to reweighting based methods (GR, CR), subsampling suffers from low average accuracy due to the reduced size of training samples, resulting in the degradation of overall trade-off and poor robust accuracy after scaling.

---

### Decision · Program_Chairs · 2023-01-20

**Decision:**

Reject

**Justification For Why Not Higher Score:**

N/A.

**Justification For Why Not Lower Score:**

N/A.

**Metareview: Summary, Strengths And Weaknesses:**

This paper proposes  a rescaling-based postprocessing method to control the trade-off between worst-group and average-case accuracies. The experiments show that the proposed method outperforms several existing baselines on two standard robustness datasets (CelebA, Waterbirds).However, the reviewers find that the motivation of this paper is very weak. Reviewers do not understand why the proposed method works. The reported baselines are not consistent with their official results. The datasets used in this paper is too trivial.